# Front-Loaded Robust Conformal Prediction:
# Heavy Calibration, Minimal Test-Time Cost

**Soroush H. Zargarbashi** [1]  **Mohammad Sadegh Akhondzadeh** [2]  **Aleksandar Bojchevski** [2]

## Abstract

Robust conformal prediction (RCP) extends conformal prediction (CP) to noisy inputs by producing prediction sets with guaranteed coverage, ensuring that the true label is contained in the set with a user-specified probability even under worst-case perturbations. Recent works use randomized smoothing, as it provides robustness for black-box models at larger radii. Currently, there exist two setups for smoothing-based RCP: one requires extensive Monte Carlo sampling at calibration and test time but results in smaller prediction sets; the other setup produces larger prediction sets but uses a single sample at both stages. In deployment, calibration—as a one-time pre-processing step—can accommodate substantially higher computational overhead than inference. Inspired by this observation, we introduce an RCP framework that strikes a balance between the two extremes of this trade-off: we increase the sample rate at calibration time while keeping it either one or very low during test time. This calibration-time sampling opens the possibility of reducing the size of the prediction sets. In production, where the number of test predictions typically far exceeds the size of the calibration set, our Front-Loaded RCP matches the computational complexity of the state of the art while producing considerably smaller prediction sets at larger radii.

## 1. Introduction

Quantification of prediction's uncertainty is crucial in safety-critical domains, and the model's confidence is not a reliable source for it (Guo et al., 2017). Moreover, existing uncertainty quantification approaches typically incur substantial computational overhead and they also lack strong statistical guarantees. These computational costs arise both during training and at inference time, often requiring retraining and multiple forward passes. Instead, conformal prediction (CP, Vovk et al. (2005); Angelopoulos et al. (2024)) wraps around any black-box model and returns prediction *sets* that are guaranteed to cover the true answer with $1-\alpha$ adjustable probability. This guarantee requires a holdout calibration set and a score function that captures the agreement between the data and any potential answer (e.g. between each image and possible target labels in image classification). During the calibration (pre-processing) step, CP computes an acceptance threshold from the calibration set; during inference, it returns the prediction set consisting of all answers whose scores exceed this threshold. Assuming exchangeability between the test and the calibration points, these prediction sets cover the true answer (or label) with $1 - \alpha$ probability.

In real-world, test inputs are often subject to noise or adversarial perturbations. Machine learning models can exhibit markedly different predictions within a small neighborhood of an input, even when the perturbations are imperceptible to humans (Goodfellow et al., 2015). Consequently, an adversary can exploit this sensitivity to alter the model's prediction without noticeably changing the input. By moving away (even unnoticeably) from the distribution of clean inputs, the exchangeability and coverage guarantee break severely towards zero (Gendler et al., 2021). Bounded-norm perturbation balls are the standard (common) model of imperceptible perturbations. Robust CP (RCP) integrates adversarial robustness techniques into conformal prediction to guarantee coverage under worst-case perturbations. In both vanilla and robust CP the coverage probability is guaranteed, therefore the race is over other properties of the sets; e.g. average set size (usability).

Usually, robustness is achieved by bounding the worst-case change in the conformity score under input perturbations, either by certifying the stability of a given score function or by introducing a new score that varies more smoothly and can be bounded more effectively. A promising approach for it is to use randomized smoothing (Kumar et al., 2020): regardless of the base model, adding random noise to the input, and defining the score over the push-forward distribution

---

[1]CISPA Helmholtz Center for Information Security [2]University of Cologne. Correspondence to: Soroush H. Zargarbashi <zargarbashi@cs.uni-koeln.de>.

*Proceedings of the $43^{rd}$ International Conference on Machine Learning*, Seoul, South Korea. PMLR 306, 2026. Copyright 2026 by the author(s).

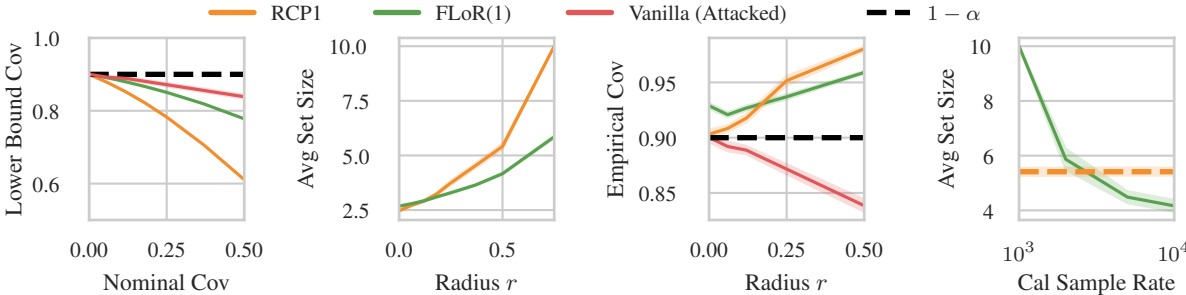

*Figure 1.* [Left] Worst-case empirical coverage (under adversarial attack) compared with the guaranteed lower bound coverage from RCP1 and FLoR[1] (without accounting for worst-case noise, see § 5). [Middle left] Average set size of BinCP, RCP1, and FLoR[1] across radii. [Middle right] Empirical coverage of methods under adversarial attacks. [Right] Average set size across various MC sampling budgets. Note that here RCP1 works with a single sample in both calibration and test (see § 5 for details).

results in a smooth score. It works with any model through black-box access, it can be defined for various perturbation schemes (Yang et al., 2020), and its prediction sets remain usable for large radii. A downside is that it often requires an extensive Monte-Carlo sampling since it relies on estimating statistics (e.g. mean) of the score around the inputs (Yan et al., 2024). This substantially limits the practicality of smoothing RCP in real-world deployments, primarily because test-time inference is computationally expensive; calibration also suffers from the same inefficiency.

While defining a new score as the mean or quantile of the push-forward distribution results in small RCP sets (Zargarbashi & Bojchevski, 2025), the computational cost remains an issue. Interestingly, Zargarbashi et al. (2025) (hereafter RCP1) show that smoothing can be integrated into CP while still requiring only a single forward pass per input (in both calibration and test). The score function is defined by a single random augmentation and the key idea is to apply certified bounds not on the scores, but directly on the coverage probabilities. By showing the convexity of the smoothing-based certified lower-bound, they circumvent the need to estimate any intermediate statistics and instead directly certify the known target probability $1 - \alpha$ (technical details are in § 3). While this test-time speedup makes the method applicable to real-time settings, the average prediction set size cannot be further reduced. Consequently, when a higher computational budget is available, RCP1 falls behind state-of-the-art sampling-heavy robust CP methods in terms of set size, without a way to improve. With the cost to set size trade-off, the open problem is to ask if there is a way to keep the *test-time computational efficiency* while decreasing the set size.

We propose FLoR[1] that bridges between the two endpoints of the mentioned trade-off following this argument: Calibration (in many settings) is a one-time pre-process. Therefore, while the test-time inference is important to be efficient, the computational overhead in this phase can be tolerated. In other words, after processing a number of test points, this

computational overhead becomes negligible in total cost as shown in Fig. 2. This led to our proposed front loaded RCP (FLoR[1]) that uses higher computational power during pre-processing calibration to attain smaller robust set size with the same single inference. Through exchangeability, we show that the coverage probability in calibration points can represent the same statistic over test points; and with that, we use conformal risk control to estimate the worst-case expected coverage by averaging over worst-case coverage probability of calibration points. Intuitively while RCP1 uses the convexity of the certified lower bound to directly apply it over the known value $1 - \alpha$ (Jensen's inequality over the coverage probability of the future test points, as mentioned in § 3), we estimate them from calibration points through Monte Carlo (MC) sampling. Except at very small radii, our method consistently outperforms RCP1; intuitively, because RCP1 relies on Jensen's inequality, which introduces additional conservatism and hence leads to larger prediction sets (see Fig. 1-right). Like RCP1, our methods apply to any binary classification certificate and extend directly to regression (see § E). Same as RCP1, FLoR[1] has the same computational complexity as verification / Lipschitz-based RCPs (see § C), while providing smaller sets that are robust to one order of magnitude larger radii.

Another limitation of RCP1 is that they return inherently stochastic prediction sets (Zargarbashi et al., 2025). For some test points the method may unluckily produce excessively large prediction sets; notably, resampling the solution will invalidate the theoretical guarantee. This issue stems from augmentation. Majority voting can be promising to reduce variance, however, it also invalidates the guarantee in RCP1, and FLoR[1]. Therefore we propose FLoR[k] which directly accounts for test time majority voting (during calibration), and returns more deterministic sets under a very low test-time sample rate. In this setup (high calibration and very low test-time sample rate), FLoR[k] also outperforms BinCP (SOTA). The code is available on Github.

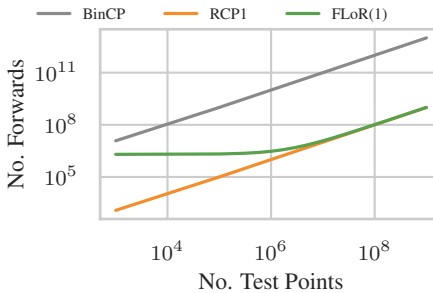

*Figure 2.* Total number of forward passes required by each method while increasing the number of expected test points. Note that both axis are log-scaled which means that parallel lines differ by exponential factor. FLoR[1] in calibration, and BinCP in both phases use $10^3$ samples per point, and 500 calibration points.

## 2. Background

**Conformal Prediction.** With $\mathcal{D} = \{\boldsymbol{x}_i, y_i\}_{i=1}^{n+1}$ as an exchangeable dataset sampled from the data distribution $\mathcal{P}_{\mathcal{X} \times \mathcal{Y}}$, and $\mathcal{D}_n$ as the calibration set taking the first $n$ data points, for clean $\boldsymbol{x}_{n+1}$, Vovk et al. (2005) show that the set $\mathcal{C}_0(\boldsymbol{x}_{n+1}) = \{y : s(\boldsymbol{x}_{n+1}, y) \geq 1 - \lambda\}$ returns the following guarantee:

$$\Pr_{\mathcal{D}}[y_{n+1} \in \mathcal{C}_0(\boldsymbol{x}_{n+1})] \geq 1 - \alpha \tag{1}$$

For any score function $s : \mathcal{X} \times \mathcal{Y} \to \mathbb{R}$ capturing the agreement between $\boldsymbol{x}$, and $y$[1], and the threshold $\lambda = 1 - \mathbb{Q}(\alpha; \{s(\boldsymbol{x}_i, y_i) : (\boldsymbol{x}_i, y_i) \in \mathcal{D}_n\})$, where $\mathbb{Q}(\beta; \mathcal{A})$ denotes the $\lfloor \alpha - \frac{1}{n} \rfloor$ quantile of the set $\mathcal{A}$. The guarantee holds regardless of the choice of score function. The choice of score function reflects in secondary properties such as smaller average set size (usability). For classification, $s$ can be the softmax outputs of the model.

**CP robust to a threat model.** The guarantee in Eq. 1 relies on the exchangeability between the calibration $\mathcal{D}_n$ and test point $\boldsymbol{x}_{n+1}$. An adversary (or natural noise) can break this exchangeability by adding imperceptible noise to the test point (evasion). Even a small perturbation can drastically decrease the coverage guarantee. Here we model the worst-case bounded noise (which we call adversarial perturbation) as the point with most miscoverage probability to be within a ball $\mathcal{B} : \mathcal{X} \to 2^{\mathcal{X}}$ around the clean input. While our approach is independent of the choice of $\mathcal{B}$, a common choice for images is the $\ell_2$ norm ball: $\mathcal{B}_r(\boldsymbol{x}) = \{\tilde{\boldsymbol{x}} : \|\tilde{\boldsymbol{x}} - \boldsymbol{x}\|_2 \leq r\}$ where $r$ is the radius of the perturbation. Robust conformal prediction (RCP) aims to extend the guarantee in

Eq. 1 to any possible perturbation within $\mathcal{B}$:

$$\mathbb{E}_{\mathcal{D}}\Big[ \min_{\tilde{\boldsymbol{x}}_{n+1} \in \mathcal{B}_r(\boldsymbol{x}_{n+1})} \Pr_u[y_{n+1} \in \mathcal{C}_r(\tilde{\boldsymbol{x}}_{n+1}; u)]\Big] \tag{2}$$
$$\geq \Pr_{\mathcal{D}, u}[y_{n+1} \in \mathcal{C}_0(\boldsymbol{x}_{n+1}; u)] \geq 1 - \alpha$$

Here $u$ encodes any randomness in defining the prediction set, i.e., randomness in the score function. For deterministic sets (i.e., no randomness in $u$), the inner probability is either 0 or 1; therefore, the guarantee in Eq. 2 reduces to:

$$\Pr_{\mathcal{D}}[y_{n+1} \in \mathcal{C}_{\mathcal{B}}(\tilde{\boldsymbol{x}}_{n+1}), \forall \tilde{\boldsymbol{x}}_{n+1} \in \mathcal{B}(\boldsymbol{x}_{n+1})] \geq 1 - \alpha$$

We use $\mathcal{C}_0$ and $\mathcal{C}_{\mathcal{B}}$ (or $\mathcal{C}_r$ for $\ell_2$ balls) to denote the vanilla and robust prediction sets, respectively. One way to attain robustness is to bound the changes in the score function within $\mathcal{B}$ — either by finding bounds for a given score function (through verification) or by designing a score function that changes slowly around the input (e.g., by averaging the score over random noise augmentations). Through randomized smoothing, Gendler et al. (2021) (RSCP) propose using the smooth score $\hat{s}(\boldsymbol{x}, y) = \mathbb{E}_\epsilon[s(\boldsymbol{x} + \epsilon, y)]$, where $\epsilon$ is random noise drawn from a standard distribution (e.g., for images, one choice is isotropic Gaussian noise). Furthermore, Zargarbashi & Bojchevski (2025) (BinCP) use a quantile-based score $\hat{s}_q(\boldsymbol{x}, y) = \mathbb{Q}(\tau; s(\boldsymbol{x} + \epsilon, y))$ for some arbitrary $\tau \in (0, 1)$ to reduce the required sample rate: methods that define the score as a statistic over the smoothed original score require extensive Monte-Carlo sampling to estimate that statistic (mean, quantile, etc.), followed by a finite-sample correction; in the case of BinCP, the required sample rate is an order of magnitude lower by bounding over a Bernoulli variable instead of a continuous variable.

Typically, we define a robust CP based on certified bounds: the lower bound $c^{\downarrow}(f, \mathcal{B}(\boldsymbol{x})) \leq \inf\{f(\boldsymbol{z}) : \boldsymbol{z} \in \mathcal{B}(\boldsymbol{x})\}$, and similarly $c^{\uparrow}(\cdot, \cdot)$ as the upper bound (with sup); we further discuss how to compute them using randomized smoothing. By definition, for all $\tilde{\boldsymbol{x}} \in \mathcal{B}(\boldsymbol{x})$, we have $c^{\downarrow}(s(\cdot, y), \mathcal{B}(\boldsymbol{x})) \leq s(\tilde{\boldsymbol{x}}, y) \leq c^{\uparrow}(s(\cdot, y), \mathcal{B}(\boldsymbol{x}))$. As an example for robust CP, Zargarbashi et al. (2024) define robust sets via test-time upper bounds over the score function : $\mathcal{C}_{\mathcal{B}}(\boldsymbol{x}_{n+1}) = \{y : c^{\uparrow}(s(\cdot, y_{n+1}), \mathcal{B}^{-1}(\boldsymbol{x}_{n+1})) \geq 1 - \lambda\}$. Here $\mathcal{B}^{-1}$ is the smallest set that contains the clean $\boldsymbol{x}$ when centered at any perturbed point $\tilde{\boldsymbol{x}} \in \mathcal{B}(\boldsymbol{x})$. For symmetric balls like $\ell_p$ we have $\mathcal{B}^{-1} = \mathcal{B}$. Another example is to set the score over randomly augmented input $\hat{s}(\boldsymbol{x}, y) = s(\boldsymbol{x} + \boldsymbol{\epsilon}, y)$ and apply the certified lower bound directly on the coverage probability instead of the score function; i.e. $\beta := \Pr[s(\boldsymbol{x} + \boldsymbol{\epsilon}, y) \geq q_\alpha]$ (Zargarbashi et al. (2025)-RCP1).

Two key building blocks underlying our methods are conformal risk control, which provides the theoretical coverage guarantee, and randomized smoothing, which yields certified upper and lower bounds. We briefly review both below.

---

[1]CP can equivalently be defined using non-conformity scores. Setups are equivalent by flipping the sign. Following prior works, we adopt the conformity formulation, as it aligns more naturally with intuition for classification.

**Conformal risk control.** The risk in conformal prediction is the exclusion of the true label. CRC (Angelopoulos et al., 2022) generalizes this guarantee to the *expected value* of any arbitrary bounded, monotone loss function $L_i(\lambda)$. For instance, in this work we set it to the miscoverage probability of each augmented point.

**Theorem 2.1** (Conformal Risk Control - rephrased). *Let $\lambda$ be a parameter (larger $\lambda$ yields more conservative output), and $L_i : \Lambda \to (-\infty, B]$ for $i = 1, \ldots, n+1$ be exchangeable random functions. If (i) $L_i$s are non-increasing right-continuous w.r.t. $\lambda$, (ii) for $\lambda_{\max} = \sup \Lambda$ we have $L_i(\lambda_{\max}) \leq \alpha$, and (iii) $\sup_\lambda L_i \leq B < \infty$, then we have:*

$$\mathbb{E}_{\mathcal{D}}[L_{n+1}(\hat{\lambda})] \leq \alpha \quad (3)$$

$$for \quad \hat{\lambda} = \inf\left\{ \lambda : \frac{\sum_{i=1}^{n} L_i(\lambda)}{n+1} + \frac{B}{n+1} \leq \alpha \right\}$$

Conformal prediction itself is a special case of conformal risk control. In case that $B = 1$, by simplifying Eq. 3, we have $\hat{\lambda} = \inf\{\lambda : \sum_{i=1}^{n} L_i(\lambda) \leq \alpha(n+1) - 1\}$.

**Bounds from randomized smoothing.** Smoothing allows us to compute certified bounds for any black-box (model or score) function. A smoothing scheme $\psi : \mathcal{X} \to \mathcal{X}$ adds random noise (from a prespecified distribution) to the input — mapping it to a random nearby point. Intuitively, in the smooth classifier (or in our case, the randomized classifier), there is a large overlap between the push forward distributions for any two inputs within $\mathcal{B}$. This overlap enables to compute closed-form upper, or lower bounds regardless of the structure of the model, and the point $\boldsymbol{x}$. In almost all cases, internally the bound is found through optimization for any classifier that shares same statistics (e.g. mean) as the black-box function $f$, and for any pair of points that are $r$ apart. As the bounds are independent of the value $\boldsymbol{x}$, and the function $f$. They are only function of the probability $p := \Pr_{\boldsymbol{\epsilon}}[f(\boldsymbol{x} + \boldsymbol{\epsilon}) = 1]$, therefore, we use the alternative notation $\mathrm{c}^{\downarrow}(p, \mathcal{B})$ instead of $\mathrm{c}^{\downarrow}(f, \mathcal{B}(\boldsymbol{x}))$.

$$\mathrm{c}^{\downarrow}(\beta, \mathcal{B}_r) \leq \min_{\tilde{\boldsymbol{x}} \in \mathcal{B}(\boldsymbol{x})} \Pr_{\boldsymbol{\epsilon}}[s(\tilde{\boldsymbol{x}} + \boldsymbol{\epsilon}) = 1] \quad (4)$$

$$\text{given} \quad \Pr_{\boldsymbol{\epsilon}}[s(\boldsymbol{x} + \boldsymbol{\epsilon}) = 1] = \beta$$

One common smoothing setup is isotropic Gaussian smoothing and $\ell_2$ ball. For example, in case of image classification, we add $\boldsymbol{\epsilon} \sim \mathcal{N}(\boldsymbol{0}, \sigma\boldsymbol{I})$ to each input $\boldsymbol{x}$. and the lower bound has a closed form solution of $\mathrm{c}^{\downarrow}(p, \mathcal{B}) = \Phi_\sigma(\Phi_\sigma^{-1}(p) - r)$ where $\Phi_\sigma$ is the CDF of the Gaussian distribution $\mathcal{N}(0, \sigma)$. Similarly, the upper bound $\mathrm{c}^{\uparrow}(\beta, \mathcal{B}_r)$ can be defined by replacing the $\min$ with $\max$ in Eq. 4 and in this specific setup the closed form is the same up to changing $-r$ with $+r$. Furthermore, for any smoothing function and threat model, analogous certified bounds can be derived following Yang et al. (2020), through the probability-bound formulation re-derived by Zargarbashi et al. (2025).

## 3. Calibration-Intensive Robust CP

We first recall RCP1 as it shares a similar logic.

**Recall: RCP1 algorithm.** RCP1 follows the standard conformal prediction pipeline with a single modification: during both calibration and inference, the input to the score function is augmented with random noise $\boldsymbol{\epsilon}$; i.e. $\hat{s}(\boldsymbol{x}, y) = s(\boldsymbol{x} + \boldsymbol{\epsilon}, y)$. The i.i.d. added noise does not break the exchangeability, and therefore the coverage guarantee is $1 - \alpha$ for clean inputs (in expectation). This probability reduces to at least $\mathrm{c}^{\downarrow}(1 - \alpha, \mathcal{B})$ under worst-case perturbations within $\mathcal{B}$ (Zargarbashi et al., 2025). To achieve a worst-case coverage of $1 - \alpha$, calibration is performed at level $\tilde{\alpha}$ such that $\mathrm{c}^{\downarrow}(1 - \tilde{\alpha}, \mathcal{B}) = 1 - \alpha$. In many cases, we can equivalently set $1 - \tilde{\alpha} = \mathrm{c}^{\uparrow}(1 - \alpha, \mathcal{B}^{-1})$.

Under the hood, RCP1 aims to certify the lower bound for $\tilde{\beta}_{n+1} = \Pr_{\boldsymbol{\epsilon}}[y_{n+1} \in \mathcal{C}_{\mathcal{B}}(\tilde{\boldsymbol{x}}_{n+1})] \geq \mathrm{c}^{\downarrow}(\beta_{n+1}, \mathcal{B})$. However, since $\beta_{n+1}$ is unknown and the method aims to avoid Monte-Carlo estimation, it leverages the convexity of the certified lower bound function to directly apply the bound to $1 - \alpha$ which is the expected value of $\beta_{n+1}$.

**RCP1 is unnecessarily conservative.** While the convexity argument (i.e., $\mathrm{c}^{\downarrow}(1 - \alpha, \mathcal{B}) \leq \mathbb{E}_{\mathcal{D}}[\mathrm{c}^{\downarrow}(\beta_{n+1}, \mathcal{B})]$) reduces the computational cost at both calibration and test time, it makes the guarantee very conservative – there is a considerable difference between $\mathbb{E}_{\mathcal{D}}[\mathrm{c}^{\downarrow}(\beta_{n+1}, \mathcal{B})]$, and $\mathrm{c}^{\downarrow}(\mathbb{E}_{\mathcal{D}}[\beta_{n+1}], \mathcal{B})$ which is also empirically shown in Fig. 1-left. On the other hand, computational cost is considerably less critical during calibration, since it is a one-time preprocessing step, whereas the dominant cost arises at inference time due to the potentially unbounded number of test examples. We propose to directly estimate $\mathbb{E}_{\mathcal{D}}[\mathrm{c}^{\downarrow}(\beta_{n+1}, \mathcal{B})]$ via Monte Carlo sampling at calibration time. Importantly, our method still requires only a single sample at inference time, thereby keeping the dominant computational cost low. In Fig. 2, we compare the cost RCP1 and FLoR[(1)] in number of required forward passes. It clearly shows that the computation costs of the two methods converge by increasing the number of test points – the bias in the computational cost becomes negligible over time.

**FLoR[(1)].** First we provide a high level view of the proof. Following Zargarbashi et al. (2025), the noise augmentation over inputs $\hat{s}(\boldsymbol{x}, y) = s(\boldsymbol{x} + \boldsymbol{\epsilon}, y)$ does not break the exchangeability and therefore, running vanilla CP over the new scores $\hat{s}$ results in $1 - \alpha$ coverage. As we are introducing randomness adding $\boldsymbol{\epsilon}$, the coverage of each point becomes a Bernoulli random variable (dependent to $\boldsymbol{\epsilon}$) with success probability $\beta_{n+1} = \Pr_{\boldsymbol{\epsilon}}[s(\boldsymbol{x}_{n+1} + \boldsymbol{\epsilon}, y_{n+1}) \geq 1 - \lambda]$, rather than a deterministic event (note that in some score functions like APS, this variable is already non-deterministic). Moving from $\boldsymbol{x}_{n+1}$ to $\tilde{\boldsymbol{x}}_{n+1} \in \mathcal{B}(\boldsymbol{x}_{n+1})$, the worst-case coverage for $\tilde{\boldsymbol{x}}_{n+1}$ bounded by $\tilde{\beta}_{n+1} = \mathrm{c}^{\downarrow}(\beta_{n+1}, \mathcal{B})$.

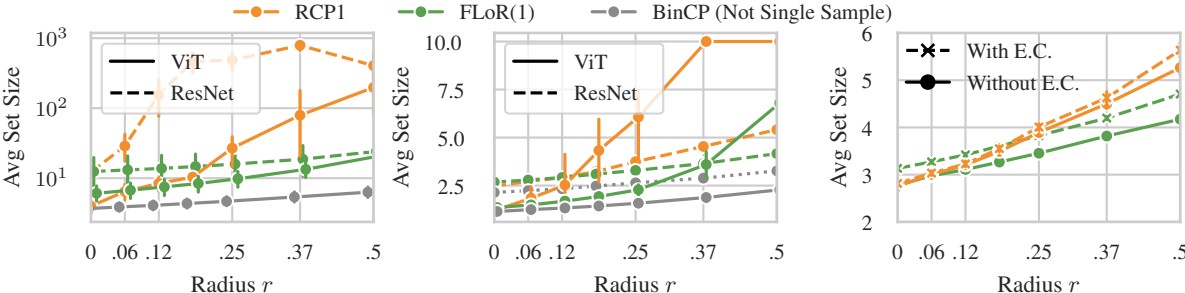

*Figure 3.* Set size across various radii for the [Left] ImageNet, and [middle] CIFAR-10 dataset both with $\sigma = 0.5$, over both ViT, and ResNet models. BinCP is shown as an ideal baseline (not comparable because of test-time MC sampling). [Right] Results for adaptive prediction set (APS) with and without finite sample correction (ResNet).

As we avoid estimating $\beta_{n+1}$ (since we aim for single sample inference), we apply conformal risk control. Let

$$L_i(\lambda) = \max_{\tilde{\boldsymbol{x}}_i \in \mathcal{B}(\boldsymbol{x}_i)} \Pr_{\boldsymbol{\epsilon}}[s(\tilde{\boldsymbol{x}}_i + \boldsymbol{\epsilon}, y_i) \leq 1 - \lambda] \quad (5)$$

Here $L_i(\lambda)$ is the risk function capturing the probability of miscoverage for the worst $\tilde{\boldsymbol{x}}_i$ with the threshold $1 - \lambda$. From the certified lower bound function we have $1 - c^\downarrow(\beta_i(\lambda), \mathcal{B}) \geq L_i(\lambda)$. Note that $\beta_i$ is implicitly a function of $\lambda$. Replacing the risk $L_i(\lambda)$ with $1 - c^\downarrow(\beta_i(\lambda), \mathcal{B})$ only increases the value $\lambda$ making the setup more conservative. Through conformal risk control (as we show further) we find a $\lambda^\star$ that guarantees $\mathbb{E}_{\mathcal{D}}[\tilde{\beta}_{n+1}] \geq 1 - \alpha$. Sampling one instance from each realization of $\tilde{\beta}_{n+1}$ (over the test points) results in expected worst-case coverage of $1 - \alpha$.

To lower bound $\tilde{\beta}_i$, we first estimate $\beta_i$ via Monte Carlo sampling. Using $m$ samples per input $\boldsymbol{x}_i$, we compute $\hat{\beta}_i = \frac{1}{m} \sum_{j=1}^{m} \mathbb{I}[s(\boldsymbol{x}_i + \boldsymbol{\epsilon}_j, y_i) \geq 1 - \lambda]$. Via Clopper-Pearson inequality (Clopper & Pearson, 1934), we obtain a lower confidence bound $\underline{\beta}_i$ such that $\Pr[\beta_i \geq \underline{\beta}_i] \geq 1 - \delta/|\mathcal{D}_n|$, which implies a union bound for success probability to be $1 - \delta$ over all calibration points. Under worst-case perturbations within $\mathcal{B}$, the coverage probability at $\tilde{\boldsymbol{x}}_i$ is lower bounded by $c^\downarrow(\underline{\beta}_i, \mathcal{B}) \leq \tilde{\beta}_i$. By replacing $1 - c^\downarrow(\underline{\beta}_i, \mathcal{B})$ with $L_i(\lambda)$ and tuning for $1 - \alpha + \delta$ coverage, we can secure the worst case $1 - \alpha$ coverage at test time.

All the above arguments lead to the following result:

**Proposition 3.1.** *Given clean calibration points $Z_1, \ldots, Z_n$ ($Z_i = (X_i, Y_i)$), and a potentially perturbed $\tilde{Z}_{n+1} = (\tilde{X}_{n+1}, Y_{n+1})$, for $\tilde{X}_{n+1} \in \mathcal{B}(X_{n+1})$, let $E_i : i \in [n+1] \sim \psi$ from a predefined smoothing scheme $\psi$, we have*

$$\Pr_{E_{n+1}, \mathcal{D}}[Y_{n+1} \in \mathcal{C}(\tilde{X}_{n+1})] \geq 1 - \alpha$$

*For $\mathcal{C}(\tilde{X}_{n+1}) = \{y : s(\tilde{X}_{n+1} + E_{n+1}, y) \geq 1 - \lambda\}$ where $1 - \lambda$ is defined as following: Let $\beta_i(\lambda) = \Pr_{E_i}[s(X_i + E_i, Y_i) \geq 1 - \lambda]$ and $\underline{\beta}_i \leq \beta_i$ with $1 - \delta/|\mathcal{D}_n|$ probability.*

*Then*

$$\lambda = 1 - \sup\{\lambda' : \frac{1}{n+1} \sum_{i=1}^{n} c^\downarrow(\underline{\beta}_i(\lambda'), \mathcal{B}) \geq 1 - \alpha + \delta\}$$

**Finding $\lambda$ in practice.** In general, for conformal risk control, the variable $\lambda$ either admits a closed-form solution (like the special case of conformal prediction) or can be computed via binary search. In our setup, where $\beta_i(\lambda)$ must be estimated from MC-samples, a naive approach would be to resample all calibration points at each of the $\mathcal{O}(\log n)$ steps of the binary search – we cannot reuse the same samples across iterations, as this corresponds to multiple hypothesis testing over $\beta_i(\lambda)$ for different values of $\lambda$. To circumvent this computational cost, we split our sampling budget into a certification and a considerably smaller tuning budget; i.e. $m = m_{\text{cert}} + m_{\text{tune}}$. Importantly, Proposition 3.1 already provides the complete formal guarantee: any $\lambda$ satisfying the constraint is valid, and the finite-sample correction is handled by the Clopper-Pearson lower bound with confidence level $\delta$. The role of the tuning step is purely computational: it is a *cheap proxy* for the binary search over $\lambda$ that Proposition 3.1 requires. Since no statistical claims are made during tuning (the same sample is reused across the whole search), no additional multiple-testing correction is needed. The statistical validity of the procedure depends *entirely* on the final validation step, which is a single run of the constraint in Proposition 3.1 using the independent $m_{\text{cert}}$ sample. This mirrors standard practice in randomized smoothing (Cohen et al., 2019). Therefore, we first compute $\lambda$ by running binary search using $m_{\text{tune}}$ samples, and then validate the resulting candidate using the $m_{\text{cert}}$ budget. During the tuning step, we ensure that the selected $\lambda$ ultimately results in a risk below $\alpha$ when validated over the larger sample rate. In other words, during tuning, we account for finite sample correction, and the certified lower bounds.

During binary search, for any trial threshold $\lambda$, we estimate the coverage probability per each calibration point $\hat{\beta}_i = \frac{1}{m_{\text{tune}}} \sum_{j=1}^{m_{\text{tune}}} \mathbb{I}[s(\boldsymbol{x}_i + \boldsymbol{\epsilon}_j, y_i) \geq 1 - \lambda]$, and compute

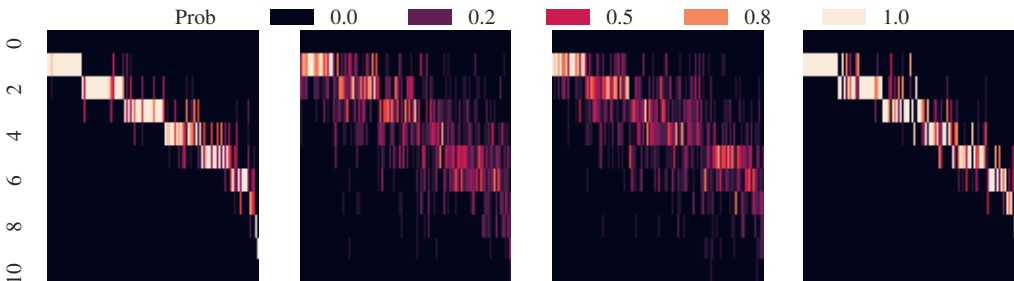

*Figure 4.* Distribution of set sizes across inputs. The x-axis shows various inputs, and the y-axis shows different set sizes. [From left to right] plots are depicting BinCP, RCP1, FLoR$^{(1)}$, and FLoR$^{(k)}$ all over the same evaluation subset of test points of CIFAR-10 dataset. Here $r = 0.25$, $\sigma = 0.5$, and we used ResNet model. In all plots, inputs are sorted with the same reference. Here BinCP and FLoR$^{(k)}$ are both calibrated with $10^4$ MC samples, and tested with 101 samples.

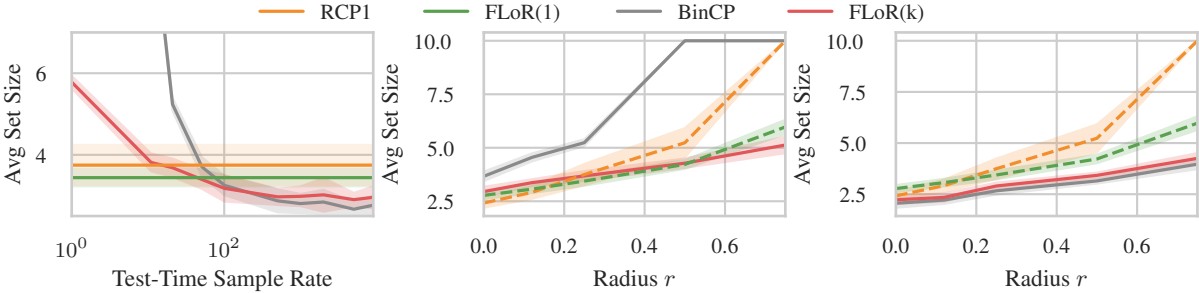

*Figure 5.* [Left] Average set size as a function of test-time sample rate at fixed radius $r = 0.25$; RCP1 and FLoR$^{(1)}$ serve as reference only, [Middle / Right] Average set size versus perturbation radius $r$ at low (21) and high (5001) test-time samples, respectively; dashed lines show the oracle baselines (RCP1, FLoR$^{(1)}$) for reference.

the confidence interval lower bound $\underline{\beta}_i$ through Clopper-Pearson intervals with confidence $1 - \overline{\delta}/|\mathcal{D}_n|$. Notably, as we later validate the $\lambda$, computing the confidence interval is not necessary. It is only to simulate how the interval length can affect the validation step, therefore we imitate the actual certification step, where the confidence lower bound is computed over $m_{\text{cert}}$ samples per calibration point – i.e. $\hat{\beta}_i \cdot m_{\text{cert}}$ successes over $m_{\text{cert}}$ samples. In order to ensure that the resulting $\lambda$ passes the validation phase, we also add an extra bias to the binary search by setting the coverage probability to $1 - \alpha + \delta + \delta_0$ where $\delta_0$ is a very small value accounting for potential mismatch between the simulated, and the actual confidence intervals. Finally, the decision rule for binary search is

$$1/(n+1)\sum_{i=1}^{n} \text{c}^{\downarrow}(\underline{\beta}_i, \mathcal{B}) \geq 1 - \alpha + \delta + \delta_0$$

Finding the best value for $\lambda$, we validate it by estimating $\hat{\beta}_i$ over $m_{\text{cert}}$ samples, and computing the lower bound $\underline{\beta}_i$, this time using the actual number of successes. The final guaranteed coverage is computed higher than $1/(n+1)\sum_{i=1}^{n} \text{c}^{\downarrow}[\underline{\beta}_i, \mathcal{B}] - \delta$. We expect this value to be above $1 - \alpha$. In the rare event that validation does not certify the re-

quired coverage (empirically zero failures with $\delta_0 = 0.005$ and 500 tuning samples across all CIFAR-10 experiments), $\delta_0$ may be slightly increased and the process repeated. The while-loop in Algorithm 1 captures this, but in practice it executes only once.

**When and why FLoR$^{(1)}$ yields better sets.** Both RCP1 and FLoR$^{(1)}$ share the same definition of the prediction set, and the same objective for calibration. The key difference arises on how to aim for that objective. RCP1 first proves the $E_{\mathcal{D}}[\beta_{n+1}] \geq 1 - \alpha$, and uses the Jensen's inequality to directly lower bound the objective with $\text{c}^{\downarrow}(1 - \alpha, \mathcal{B})$. Instead FLoR$^{(1)}$ finds this value by approximating worst case $\tilde{\beta}_{n+1}$ from calibration points. Therefore, setting aside the practical implementation, and finite sample correction, the improvement in set size boils down to the distance of the Jensen's lower bound with the real expectation. (i.e. $f(E[X]) \leq E[f(X)]$ for convex $f$). Asymptotically, FLoR$^{(1)}$ always results in smaller prediction sets compared to RCP1. However, in practical cases, when $r$ is very small, this gap diminishes, and the combination of finite sample correction, and conservative implementation ($\delta_0$) inflates the sets w.r.t. RCP1. The following proposition and remark make this precise.

**Proposition 3.2** (Asymptotic dominance of $\text{FLoR}^{(1)}$ over RCP1). *In the asymptotic regime ($n, m \to \infty$, so finite-sample corrections vanish), let $q_{\text{RCP1}}$ and $q_{\text{FLoR}}$ denote the calibration thresholds computed by RCP1 and $\text{FLoR}^{(1)}$ respectively, each targeting robust $1 - \alpha$ coverage. Then: (i) $q_{\text{FLoR}} \geq q_{\text{RCP1}}$ for all $r \geq 0$. The inequality is strict for $r > 0$ when Jensen gap is non-zero. (ii) $|\mathcal{C}_{\text{FLoR}}(\tilde{\boldsymbol{x}})| \leq |\mathcal{C}_{\text{RCP1}}(\tilde{\boldsymbol{x}})|$ for every pair $(\tilde{\boldsymbol{x}}, \boldsymbol{\epsilon})$.*

*Proof.* Both methods define $\mathcal{C}(\tilde{\boldsymbol{x}}) = \{y : s(\tilde{\boldsymbol{x}} + \boldsymbol{\epsilon}, y) \geq q\}$; a higher threshold $q$ yields a smaller set. RCP1 finds the supremum of $q$ such that $c^{\downarrow}(\bar{\beta}(q), \mathcal{B}) \geq 1 - \alpha$, where $\bar{\beta}(q) = \mathbb{E}_{\mathcal{D}}[\beta_{n+1}(q)]$. $\text{FLoR}^{(1)}$ finds the supremum of $q$ such that $\mathbb{E}_{\mathcal{D}}[c^{\downarrow}(\beta_{n+1}(q), \mathcal{B})] \geq 1 - \alpha$. Since $p \mapsto c^{\downarrow}(p, \mathcal{B})$ is convex, Jensen's inequality gives

$$\mathbb{E}[c^{\downarrow}(\beta, \mathcal{B})] \geq c^{\downarrow}(\mathbb{E}[\beta], \mathcal{B}) \quad \text{for all } q.$$

Hence the feasible set for $\text{FLoR}^{(1)}$ is a superset of that for RCP1, so $q_{\text{FLoR}} \geq q_{\text{RCP1}}$. Part (ii) follows because $q_{\text{FLoR}} \geq q_{\text{RCP1}}$ implies $\{y : s \geq q_{\text{FLoR}}\} \subseteq \{y : s \geq q_{\text{RCP1}}\}$. $\square$

*Remark* 3.3 (Quantifying the Jensen gap for Gaussian smoothing). The gap $\mathbb{E}[c^{\downarrow}(\beta, \mathcal{B})] - c^{\downarrow}(\bar{\beta}, \mathcal{B})$ governs how much more conservative RCP1 is than $\text{FLoR}^{(1)}$. By a second-order Taylor expansion of $c^{\downarrow}(\cdot, \mathcal{B})$ around $\bar{\beta}$:

$$\mathbb{E}[c^{\downarrow}(\beta, \mathcal{B})] - c^{\downarrow}(\bar{\beta}, \mathcal{B}) \approx \tfrac{1}{2} c^{\downarrow''}(\bar{\beta}, \mathcal{B}) \text{Var}[\beta]. \quad (6)$$

For Gaussian smoothing ($\boldsymbol{\epsilon} \sim \mathcal{N}(\boldsymbol{0}, \sigma^2 \boldsymbol{I})$) and $\ell_2$ balls, $c^{\downarrow}(p, r) = \Phi_{\sigma}(\Phi_{\sigma}^{-1}(p) - r)$, and direct differentiation gives

$$c^{\downarrow''}(p, r) = \frac{r}{\sigma} \cdot \frac{\phi(\Phi^{-1}(p) - r/\sigma)}{\phi(\Phi^{-1}(p))^2}, \quad (7)$$

where $\phi$ and $\Phi$ are the standard Gaussian PDF and CDF. This expression is zero at $r = 0$, non-negative for all $r > 0$, and returns to zero as $r \to \infty$. Hence the Jensen gap — and the size advantage of $\text{FLoR}^{(1)}$ — is nonzero for all $r > 0$ and vanishes only at the degenerate extremes.

## 4. Reducing Stochasticity

Both RCP1 and $\text{FLoR}^{(1)}$ produce inherently stochastic prediction sets – the indicator $\mathbb{I}[s(\boldsymbol{x}_{n+1} + \boldsymbol{\epsilon}, y) \geq 1 - \lambda]$ remains random even after fixing $\boldsymbol{x}_{n+1}$, $y$, and $\lambda$. Methods such as BinCP exhibit comparatively lower variability by definition of their score function. Fig. 4 clearly shows this stochasticity reflected in the variance of prediction set sizes across inputs. As in the single sample regime this stochasticity is inevitable, we need to sample more from $\boldsymbol{x}_{n+1} + \boldsymbol{\epsilon}$ to reduce the variance. Our goal is to achieve the smallest set size using a very low test-time sample rate. One approach is to use BinCP with different sample rates during calibration

(higher) and test (lower). Alternatively, we can aggregate prediction sets from repeated sampling through different operators: *union* preserves coverage monotonicity, but it rapidly inflates the prediction set. *Majority voting* stabilizes set size but provably reduces guaranteed coverage to $1 - 2\alpha$ (Angelopoulos et al., 2024). We instead propose $\text{FLoR}^{(k)}$ through the same framework as $\text{FLoR}^{(1)}$ which is aware of test time resampling, and majority vote aggregation which results in robust coverage guarantees, and returns set sizes that are smaller than BinCP for the very-low sampling regime (e.g. 21 to 101 samples).

**$\text{FLoR}^{(k)}$.** While majority vote reduces the guarantee to $1 - 2\alpha$, we overcome this issue by calibrating with an objective aware of a majority-vote test-time aggregator. Let $\Phi_{\text{bin}}(t, k, p)$ denote the binomial CDF. Our method identifies positive values $(k, \alpha_0, \beta_0)$ that satisfy the following:

$$(1 - \alpha + \alpha_0)\left(1 - \Phi_{\text{bin}}\left(\tfrac{k-1}{2}, k, \tfrac{1}{2} + \beta_0\right)\right) \overset{?}{\geq} 1 - \alpha \quad (8)$$

Note that fixing two, the third variable always has a solution. Following proposition holds.

**Proposition 4.1.** *With an odd integer $k \geq 1$, positive offsets $(\alpha_0, \beta_0) \geq 0$ ($\beta_0 \leq \frac{1}{2}$, $\alpha_0 \leq \alpha$) satisfying Eq. 8, and a smoothing distribution $\psi$. Let $(Z_i)_{i=1}^{n+1}$ with $Z_i = (X_i, Y_i)$ be exchangeable, and let $\mathcal{B}_r(x)$ be the perturbation ball. For each $i \in [n]$ and threshold $\lambda$, define the (robust) inclusion probability*

$$\tilde{\beta}_i(\lambda) := \min_{\tilde{X}_i \in \mathcal{B}_r(X_i)} \Pr_{E_i \sim \psi}\left[s(\tilde{X}_i + E_i, Y_i) \geq 1 - \lambda\right].$$

*estimated in same way as Proposition 3.1, resulting in lower confidence bound $\underline{\beta}_i(\lambda)$ satisfying*

$$\Pr\left[\underline{\beta}_i(\lambda) \leq \beta_i(\lambda)\right] \geq 1 - \delta/|\mathcal{D}_n|.$$

*consider the risk function*

$$L_i(\lambda) := \mathbb{I}\left[c^{\downarrow}(\underline{\beta}_i(\lambda), \mathcal{B}) \leq \tfrac{1}{2} + \beta_0\right], \qquad i \in [n], \quad (9)$$

*if $\lambda$ satisfies the following inequality*

$$\frac{1}{n+1}\left(\sum_{i=1}^{n} L_i(\hat{\lambda}) + 1\right) \leq \alpha - \alpha_0 - \delta.$$

*The majority vote prediction set $\mathcal{C}^{(k)}$ defined as following*

$$\mathcal{C}^{(k)}(\boldsymbol{x}_{n+1}) = \{y : \frac{1}{k}\sum_{t=1}^{k}\mathbb{I}[s(\boldsymbol{x}_{n+1} + \boldsymbol{\epsilon}_t, y) \geq 1 - \lambda] > \frac{1}{2}\}$$

*has the robust $1 - \alpha$ guarantee.*

*Proof.* Consider the following risk function

$$L_i'(\lambda) = \mathbb{I}[\min_{\tilde{\boldsymbol{x}}_i \in \mathcal{B}(\boldsymbol{x}_i)} \Pr_{\boldsymbol{\epsilon}}[s(\tilde{\boldsymbol{x}}_i + \boldsymbol{\epsilon}, y_i) \geq 1 - \lambda] \leq \frac{1}{2} + \beta_0]$$

Same as Proposition 3.1 this risk monotonically decreasing w.r.t. $\lambda$, right continuous, and upper bounded by 1. By the definition of the certified lower bound function we have $L_i(\lambda) \geq L_i'(\lambda)$ for $L_i$ defined in Eq. 9. Therefore calibrating with $L_i$ through risk control, yields a $\lambda$ that satisfies $E_{\mathcal{D}}[L_{n+1}(\lambda)] \leq \alpha - \alpha_0$ which implies the following

$$\Pr_{\mathcal{D}}\big[\beta_{n+1}(\lambda) \geq \tfrac{1}{2} + \beta_0\big] \geq 1 - \alpha + \alpha_0.$$

Note that

$$\Pr_{\substack{\mathcal{D} \\ \boldsymbol{\epsilon}_{n+1,1:k}}} \big[y_{n+1} \in \mathcal{C}^{(k)}(\boldsymbol{x}_{n+1})\big] \geq \Pr_{\mathcal{D}}\big[\beta_{n+1}(\lambda) \geq \tfrac{1}{2} + \beta_0\big] \times$$

$$\Pr_{\boldsymbol{\epsilon}_{n+1,1:k}}\big[y_{n+1} \in \mathcal{C}^{(k)}(\boldsymbol{x}_{n+1}) \mid \beta_{n+1}(\lambda) \geq \frac{1}{2} + \beta_0\big]$$

where the second term captures the probability of majority vote over $k$ samples including the true label, while the probability of including it is higher than $\frac{1}{2} + \beta_0$. We have

$$\Pr_{\boldsymbol{\epsilon}_{n+1,1:k}}\big[y_{n+1} \in \mathcal{C}^{(k)}(\boldsymbol{x}_{n+1}) \mid \beta_{n+1}(\lambda) \geq \frac{1}{2} + \beta_0\big]$$

$$\geq \Pr_{\boldsymbol{\epsilon}_{n+1,1:k}}\Big[\mathrm{Bin}\Big(k, \tfrac{1}{2} + \beta_0\Big) \geq \tfrac{k+1}{2}\Big]$$

$$\geq \Big(1 - \Phi_{\mathrm{bin}}\Big(\tfrac{k-1}{2}, k, \tfrac{1}{2} + \beta_0\Big)\Big)$$

Therefore combining the two terms with $(k, \alpha_0, \beta_0)$ satisfying Eq. 8 we have

$$\mathbb{E}_{\mathcal{D}}\big[\Pr_{\boldsymbol{\epsilon}_{1:k}}\big(y_{n+1} \in \mathcal{C}^{(k)}(\boldsymbol{x}_{n+1})\big)\big]$$

$$\geq (1 - \alpha + \alpha_0)\Big(1 - \Phi_{\mathrm{bin}}\Big(\tfrac{k-1}{2}, k, \tfrac{1}{2} + \beta_0\Big)\Big) \geq 1 - \alpha.$$

$$\square$$

**Practical hyperparameter selection for FLoR$^{(k)}$.** The three parameters $(k, \alpha_0, \beta_0)$ are coupled through Eq. 8 but their selection is efficient in practice. $k$ is a deployment-time design choice (the number of test-time samples, e.g. batch size or concurrent inference budget); higher $k$ is always better since the inner factor in Eq. 8 is monotonically increasing in $k$ for any $\beta_0 > 0$. For fixed $k$, we recommend setting $\alpha_0 = \alpha/10$ as a default (this works well across all our experiments), and solving for $\beta_0$ via a binary search over Eq. 8, which is data-independent and runs in milliseconds. The hyper-parameter selection thus reduces to a one-dimensional search over $\alpha_0 \in (0, \alpha/2)$ with $\beta_0$ determined automatically and tuning performed on the same calibration set used for $\lambda$ selection, without any separate hold-out.

## 5. Experiments

Via empirical evaluation we show that (i) FLoR$^{(1)}$ provides robust guarantee: while theoretically guaranteed, we also empirically show that the coverage of FLoR$^{(1)}$ remains above the nominal $1 - \alpha$ in presence of adversarial attack. (ii) FLoR$^{(1)}$ returns smaller sets compared to prior state of the art RCP1: We show that as we increase the calibration sample rate, the set size in FLoR$^{(1)}$ decreases. This is while due to its nature, RCP1 can not get more efficient by increasing the computational resources. (iii) FLoR$^{(k)}$ reduces the stochasticity of the prediction set to a level equal with BinCP while under a very low sample rate, it outperforms BinCP in set size.

Our comparison baselines are RCP1 (Zargarbashi et al., 2025), and BinCP (Zargarbashi & Bojchevski, 2025). There is a tradeoff between the two methods: RCP1 works with a single noise augmented sample per input (which works with roughly the same computation time as normal model forward) while it returns relatively larger sets. On the opposite side of the trade-off, BinCP returns smaller sets while it requires extensive Monte-Carlo sampling (both at the calibration and test time, (Yan et al., 2024)) – to outperform RCP1 BinCP needs roughly between 70 - 150 MC samples per each input. FLoR$^{(1)}$ stays in the middle of this trade-off, but close to RCP1. It works with single sample at the test time (where the actual load is), while it returns smaller set size at the cost of Monte Carlo sampling only during calibration time. Note that while we report the results for BinCP in some plots, we only bring those results as a comparison to best possible setup, while with the extensive test time sampling of BinCP, *the two methods are not comparable*.

Following Zargarbashi et al. (2025), we employ two distinct classification pipelines. (i) The first is a computationally lightweight configuration: we rely on ResNet models trained with noise augmentation from Cohen et al. (2019). With the relatively small model size, operating at large sampling rates—while computationally inefficient—remains feasible in practice. (ii) We also study a second, more computationally demanding pipeline introduced by Carlini et al. (2022). Here, each input is first denoised using a diffusion model and subsequently classified by a vision transformer. In this setup it is unrealistically expensive to apply Monte-Carlo sampling during test-time, which is a shortcoming of BinCP – this setup exclusively fits RCP1, and FLoR$^{(1)}$. The calibration-time MC sampling in FLoR$^{(1)}$ runs only once prior to deployment which is a tolerable cost.

**Valid coverage guarantees.** Same as previous robust CP methods, we show that FLoR$^{(1)}$ satisfies the guarantee in Eq. 2, both theoretically (see § 3) and empirically (see Fig. 1-third from right). Both methods remain above $1 - \alpha$. Note that the adversarial attack is aware of randomized smoothing which makes it even stronger than the conventional PGD.

**Conservativeness gap.** As long as the lower bound coverage is provided, we prefer the empirical coverage to be closer to $1 - \alpha$, since increasing the coverage naturally re-

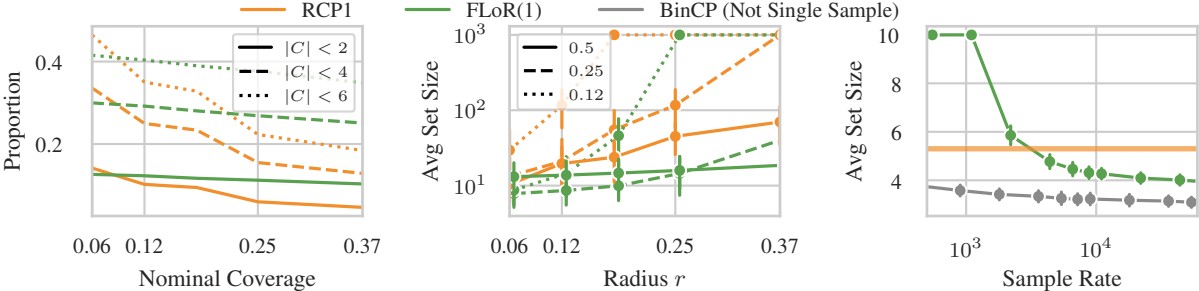

*Figure 6.* [Left] Proportion of sets with size $\leq 1$, 3, and 5, for ImageNet dataset, and ResNet model with $\sigma = 0.5$. [Middle] On the same setup, the average set size for both methods across different model and smoothing $\sigma$'s. [Right] Average set size by increasing sample rate (here $r = 0.5$), and the results are over CIFAR-10 dataset and ResNet model.

sults in inflating the prediction sets. Since RCP1 shortcuts the estimation of $\beta_{n+1}$ values (see § 3) using the convexity of certified lower-bound, it becomes unnecessary more conservative. We remedy that by directly estimating the worst case coverage from the calibration points. In Fig. 1-right, show that FLoR$^{(1)}$ is significantly less conservative and closer to the actual empirical value under adversarial attack. There we do not account for the perturbation and directly query on the worst case guaranteed coverage.

**Average Set Size.** In Fig. 3 we show that overall, FLoR$^{(1)}$ results in smaller prediction sets for both ImageNet and CIFAR-10 datasets. As we increase the radius $r$, FLoR$^{(1)}$ outperforms RCP1 by higher margins. The effect is consistent across the efficient (ResNet) and computationally expensive (diffusion + ViT) setups. Notably ViT shows a significantly better performance, for the ImageNet dataset. Intuitively we do not expect FLoR$^{(1)}$ to outperform BinCP as it also benefits from expensive MC sampling during the test time. We also report the proportion of small sets ($|\mathcal{C}(\boldsymbol{x}_{n+1})| \leq 1$, 3, and 5) for the ImageNet dataset in Fig. 6-left. For test-time sampling, we compare our FLoR$^{(k)}$ with BinCP, both calibrated with $10^4$ samples and allowed for various sample-rates from 21 to 9001. From the motivation of the work, clearly we favor the setup with very small test-time sample rate, and in that setup FLoR$^{(k)}$ outperforms BinCP. We also compare the FLoR$^{(1)}$ in set size using the adaptive prediction sets (APS) score function from Romano et al. (2020). We discuss the reasons why RCP1 performs better at smaller radii in the following.

**Discussion for smaller radii $r$.** Setting aside the finite sample correction and accounting for small discrepancies between tuning and validation phases (via $\delta_0$), by definition, FLoR$^{(1)}$ performs better than RCP1 for any radius. In practice these two factors slightly affect the set size resulting in the performance of RCP1 being better for very small radii. By increasing the radius $r$, this effect becomes negligible. To highlight this, we also compare the set size of APS score with and without these two factors

(see Fig. 3-right). Notably by disabling finite sample correction (considering asymptotically valid setup), we see that FLoR$^{(1)}$ outperforms RCP1 across all radii.

**Higher sample-rates.** In Fig. 6-right we show how increasing the calibration-time sample rate can help the set size efficiency. While reported, note that again BinCP is not comparable with FLoR$^{(1)}$, as by using the same computation for BinCP we can run FLoR$^{(1)}$ with orders of magnitude higher sample rate; e.g. $|\mathcal{D}_n| = 200$, only after inference over 50,000 test point, the computational cost of FLoR$^{(1)}$ with the highest experimented sample rate (60,000) becomes lower than BinCP showing the same set size at a significantly low sample rate (250). Therefore, here, the $x$ axis does not represent the same thing.

**Effect of $\sigma$.** Consistent with all previous smoothing-based RCP approaches, by increasing the smoothing $\sigma$, the set size slightly increases for all radii, while the slope of the inflation for larger $r$'s remains steady.

## 6. Conclusion

We introduced FLoR$^{(1)}$, a front-loaded robust conformal prediction framework: keeping the prediction time as fast as the model inference (like SOTA), by using more computational power during pre-processing (calibration) we reduce the average set size and maintain the same certified robustness guarantee. Our approach has the advantages of the both ends in the trade-off between computational efficiency (single sample RCP with relatively larger set sizes) and set-size efficiency (through expensive MC sampling). While in FLoR$^{(1)}$, and other smoothing-based single sample (RCP1) the prediction set is random, we proposed FLoR$^{(k)}$ which under a very low test-time sampling budget returns more deterministic sets still with set size better than the state of the art with same sampling budget. Notably, our methods work with any black-box model, and there are no assumptions on the model architecture, score function, or data-generating distribution. Limitations and literature review are in § C.

## Impact Statement

This work advances robust uncertainty quantification for machine-learning systems. Its main contribution is methodological: it improves the computational efficiency of robust conformal prediction while preserving formal coverage guarantees. We do not anticipate direct negative societal impacts beyond those commonly associated with deploying machine-learning models, adversarial robustness, and uncertainty quantification.

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

# A. Background on Randomized Smoothing

Randomized smoothing is a technique for certifying the robustness of any black-box model against input perturbations, without requiring access to the model's architecture, weights, or gradients (Cohen et al., 2019; Yang et al., 2020). By reasoning about the model's behavior over a distribution of randomly noisy inputs, it provides closed-form certified bounds on how much the model's output can change as the input moves within a threat model $\mathcal{B}$. We give a self-contained introduction here, starting from the high-level intuition and building up to the formal guarantees used throughout this work.

**High-level idea.** Let $f : \mathcal{X} \to \{0, 1\}$ be any binary function (e.g., the indicator that a conformity score exceeds a threshold: $f(\boldsymbol{z}) = \mathbb{I}[s(\boldsymbol{z}, y) \geq 1 - \lambda]$). A smoothing scheme $\psi$ adds random noise $\boldsymbol{\epsilon} \sim \psi$ to the input, mapping it to a random nearby point $\boldsymbol{x} + \boldsymbol{\epsilon}$. With that, we can define a *smooth* version of $f$ (the push-forward distribution:

$$g(\boldsymbol{x}) := \Pr_{\boldsymbol{\epsilon} \sim \psi}[f(\boldsymbol{x} + \boldsymbol{\epsilon}) = 1] = \mathbb{E}_{\boldsymbol{\epsilon}}[f(\boldsymbol{x} + \boldsymbol{\epsilon})].$$

Rather than evaluating $f$ at $\boldsymbol{x}$ directly, we ask with what probability $f$ outputs 1 over the distribution of noisy copies of $\boldsymbol{x}$. A common choice for continuous inputs (e.g., images) is isotropic Gaussian noise $\boldsymbol{\epsilon} \sim \mathcal{N}(\boldsymbol{0}, \sigma \boldsymbol{I})$.

Now consider a perturbed input $\tilde{\boldsymbol{x}} \in \mathcal{B}(\boldsymbol{x})$. The push-forward distributions of $\boldsymbol{x} + \boldsymbol{\epsilon}$ and $\tilde{\boldsymbol{x}} + \boldsymbol{\epsilon}$ under $\psi$ have a large overlap: the closer $\tilde{\boldsymbol{x}}$ is to $\boldsymbol{x}$, the more these distributions overlap. Crucially, this overlap is a property of the *noise distribution alone* — it does not depend on the model $f$ at all. Therefore, $g(\tilde{\boldsymbol{x}})$ cannot deviate too far from $g(\boldsymbol{x})$, and the maximum permissible deviation can be characterized in closed form for standard choices of $\psi$ and $\mathcal{B}$. This is the central insight: by marginalizing over the noise, any model becomes smooth in a quantifiable way, regardless of the internal architecture, or weights.

**Certified lower bound.** Let $\beta := g(\boldsymbol{x}) = \Pr_{\boldsymbol{\epsilon}}[f(\boldsymbol{x} + \boldsymbol{\epsilon}) = 1]$ be the smooth coverage probability at a clean input $\boldsymbol{x}$. The certified lower bound $c^{\downarrow}(\beta, \mathcal{B}_r)$ is computed such that:

$$c^{\downarrow}(\beta, \mathcal{B}_r) \leq \min_{\tilde{\boldsymbol{x}} \in \mathcal{B}_r(\boldsymbol{x})} \Pr_{\boldsymbol{\epsilon}}[f(\tilde{\boldsymbol{x}} + \boldsymbol{\epsilon}) = 1] \quad \text{given} \quad \Pr_{\boldsymbol{\epsilon}}[f(\boldsymbol{x} + \boldsymbol{\epsilon}) = 1] = \beta. \tag{10}$$

This bound does not depend on the specific model $f$ or on the point $\boldsymbol{x}$ — only on $\beta$, the noise distribution $\psi$, and the threat model $\mathcal{B}_r$. This universality follows from the fact that the tightest valid lower bound can be found by optimizing over *all* measurable binary functions $h : \mathcal{X} \to \{0, 1\}$ that share the same mean as $f$ at $\boldsymbol{x}$. Formally (Lee et al., 2019; Zargarbashi et al., 2025):

$$c^{\downarrow}(\beta, \mathcal{B}_r) = \min_{h \in \mathcal{H}} \Pr_{\boldsymbol{\epsilon}}[h(\tilde{\boldsymbol{t}} + \boldsymbol{\epsilon}) = 1] \quad \text{s.t.} \quad \Pr_{\boldsymbol{\epsilon}}[h(\boldsymbol{t} + \boldsymbol{\epsilon}) = 1] = \beta, \tag{11}$$

where $\mathcal{H}$ is the set of all measurable functions from $\mathcal{X}$ to $\{0, 1\}$, and $(\boldsymbol{t}, \tilde{\boldsymbol{t}})$ is a pair of *canonical points* with $\|\boldsymbol{t} - \tilde{\boldsymbol{t}}\|_2 = r$. For symmetric smoothing schemes and symmetric balls (such as isotropic Gaussian and $\ell_2$), the optimization in Eq. 11 is translation and rotation invariant: the bound depends only on the distance $r$ between the inputs, not on their absolute position in input space. Hence the bound is the same for any $(\boldsymbol{x}, \tilde{\boldsymbol{x}})$ pair with $\|\boldsymbol{x} - \tilde{\boldsymbol{x}}\|_2 \leq r$, and it suffices to evaluate Eq. 11 at one representative pair, e.g., $\boldsymbol{t} = \boldsymbol{0}$ and $\tilde{\boldsymbol{t}} = [r, 0, \ldots, 0]^{\top}$. Similarly, the certified upper bound $c^{\uparrow}(\beta, \mathcal{B}_r)$ is defined by replacing $\min$ with $\max$ in Eq. 11.

The solution to Eq. 11 is obtained via a Neyman–Pearson argument: the worst-case binary function $h^*$ assigns outputs based on the likelihood ratio between the noise distributions centered at $\tilde{\boldsymbol{t}}$ and $\boldsymbol{t}$. For isotropic Gaussian noise, the level sets of this ratio are half-spaces, which yields a clean closed-form solution.

**Gaussian smoothing and $\ell_2$ ball.** For isotropic Gaussian noise $\boldsymbol{\epsilon} \sim \mathcal{N}(\boldsymbol{0}, \sigma \boldsymbol{I})$ and $\ell_2$ perturbation balls $\mathcal{B}_r(\boldsymbol{x}) = \{\tilde{\boldsymbol{x}} : \|\tilde{\boldsymbol{x}} - \boldsymbol{x}\|_2 \leq r\}$, the certified lower and upper bounds have the following closed forms (Cohen et al., 2019; Lee et al., 2019):

$$c^{\downarrow}(\beta, \mathcal{B}_r) = \Phi_{\sigma}\left(\Phi_{\sigma}^{-1}(\beta) - r\right), \qquad c^{\uparrow}(\beta, \mathcal{B}_r) = \Phi_{\sigma}\left(\Phi_{\sigma}^{-1}(\beta) + r\right), \tag{12}$$

where $\Phi_{\sigma}$ is the CDF of $\mathcal{N}(0, \sigma)$.

**Monte Carlo estimation and finite-sample correction.** For any black-box model, $\beta = \Pr_{\boldsymbol{\epsilon}}[f(\boldsymbol{x} + \boldsymbol{\epsilon}) = 1]$ cannot be computed in closed form and must be estimated from samples. Drawing $m$ i.i.d. noise vectors $\{\boldsymbol{\epsilon}_j\}_{j=1}^{m} \sim \psi$, we compute the empirical estimate $\hat{\beta} = \frac{1}{m} \sum_{j=1}^{m} \mathbb{I}[f(\boldsymbol{x} + \boldsymbol{\epsilon}_j) = 1]$. Since $m\hat{\beta} \sim \text{Binomial}(m, \beta)$, the Clopper-Pearson interval (Clopper & Pearson, 1934) provides a one-sided lower confidence bound $\underline{\beta}$ satisfying $\Pr[\beta \geq \underline{\beta}] \geq 1 - \delta$. Substituting $\underline{\beta}$ into the monotone function $c^{\downarrow}(\cdot, \mathcal{B})$ yields a valid lower bound on the worst-case coverage with the same confidence. When this

---

**Algorithm 1** FLoR$^{(1)}$

---

**Require:** Calibration set $\mathcal{D}_n = \{(\boldsymbol{x}_i, y_i)\}_{i=1}^n$, potentially perturbed test point $\tilde{\boldsymbol{x}}_{n+1}$, score function $s : \mathcal{X} \times \mathcal{Y} \to \mathbb{R}$, smoothing distribution $\psi$, threat model $\mathcal{B}$, nominal miscoverage $\alpha$, confidence parameter $\delta$, total MC budget $m = m_{\text{tune}} + m_{\text{cert}}$, offset $\delta_0$
**Ensure:** Robust prediction sets $\tilde{\boldsymbol{x}}_{n+1} \to \mathcal{C}_{\mathcal{B}}(\tilde{\boldsymbol{x}}_{n+1})$ with expected coverage probability $1 - \alpha$
 1: $\lambda \leftarrow +\infty$
 2: **while** Validation$(\mathcal{D}_n, s, \psi, \mathcal{B}, \alpha, \delta, m_{\text{tune}}, \lambda)$ from Algorithm 3 returns False **do**
 3:     Compute $\lambda \leftarrow$ Tuning$(\mathcal{D}_n, s, \psi, \mathcal{B}, \alpha, \delta, m_{\text{tune}}, \delta_0)$ from Algorithm 2
 4: **end while**
 5: For a new input $\boldsymbol{x}_{n+1}$:
 6:     draw $\boldsymbol{\epsilon} \sim \psi$
 7:     set $\mathcal{C}_{\mathcal{B}}(\boldsymbol{x}_{n+1}) := \{y \in \mathcal{Y} : s(\boldsymbol{x}_{n+1} + \boldsymbol{\epsilon}, y) \geq \lambda\}$
 8: **Return** $\mathcal{C}_{\mathcal{B}}(\cdot)$

---

must hold for $n$ calibration inputs simultaneously, a union bound with per-input confidence $\delta/n$ ensures a joint failure probability of at most $\delta$.

The number of samples $m$ controls the tightness of $\beta$: larger $m$ shrinks the confidence interval but increases computational cost. Methods that estimate continuous statistics of the smooth score (e.g., the mean (Gendler et al., 2021) or quantile (Zargarbashi et al., 2024)) require $m \sim 10^4$ samples per input to achieve acceptable bounds. By contrast, using a binary certificate — certifying the probability of the threshold-exceedance event $\mathbb{I}[s(\boldsymbol{x} + \boldsymbol{\epsilon}, y) \geq 1 - \lambda]$ as a Bernoulli trial — allows a direct application of the Binomial confidence interval, reducing the required sample count by an order of magnitude (Zargarbashi & Bojchevski, 2025). Both FLoR$^{(1)}$ and FLoR$^{(k)}$ use a binary certificate.

## B. Algorithm and time complexity

Here we provide the algorithm for FLoR$^{(1)}$. The prediction set is defined in the same way as RCP1: $\mathcal{C}_{\mathcal{B}}(\boldsymbol{x}_{n+1}) := \{y \in \mathcal{Y} : s(\boldsymbol{x}_{n+1} + \boldsymbol{\epsilon}, y) \geq 1 - \lambda\}$. Notably for an easier notation, in 1 we use $\lambda$ instead of $1 - \lambda$. Additionally, while the theory is expressed in risk which is set to the miscoverage probability, here we define everything with the coverage probability. Notably this decision only changes few steps and does not affect the validity of conformal risk control; these changes are only for better readability, and by a change in sign the setup is equivalent as proposed in Proposition 3.1. We define the algorithm in each block separately, and finally Algorithm 1 combines them all together. Usually with a small $\delta_0$ the "while" loop in Algorithm 1 works only once, and we do not need to resample again.

**Tuning to find $\lambda$ (Algorithm 2).** In this step, we first find a candidate $\lambda$ that potentially results in robust $1 - \alpha$ coverage guarantee. Since the validation evaluates this quantity using a finite Monte Carlo budget, we simulate the behavior of a substantially larger sampling budget – we estimate each $\beta$ with $m_{\text{tune}}$ samples and pretend this number is estimated with $m_{\text{cert}}$ samples. This mismatch does not affect the validity of the conformal risk control guarantee, since the certification phase is later performed using an independent set of $m_{\text{cert}}$ and confidence intervals here are just a simulation to prevent failure in the next step.

**Validation (Algorithm 3).** Given a candidate $\lambda$, this step ensures that under the worst case noise this threshold still results in $1 - \alpha$ coverage probability. Notably this step is similar to one iteration of the binary search, only this time with the actual $m_{\text{cert}}$ samples.

**Computational complexity.** Table 1 shows the computational complexity of each method in both calibration and test phases. Notably since calibration is a pre-processing step, and the calibration set is small, high computational cost at this stage can be considered negligible. In almost all setups, the test set is assumed to have unlimited size, and in real-life scenarios, test-time requires a quick response where FLoR$^{(1)}$ and RCP1 align with that need. As shown in Fig. 2, as the number of test points increases, the computational costs of FLoR$^{(1)}$ and RCP1 eventually converge. Notably, this evaluation is conducted over $10^8$ test points, while in reality this number can be significantly higher.

---

**Algorithm 2** FLoR$^{(1)}$: Pre-computation (tuning samples)

---

**Require:** Calibration set $\mathcal{D}_n = \{(\boldsymbol{x}_i, y_i)\}_{i=1}^n$, score function $s : \mathcal{X} \times \mathcal{Y} \to \mathbb{R}$, smoothing distribution $\psi$, threat model $\mathcal{B}$, nominal miscoverage $\alpha$, confidence parameter $\delta$, total MC budget $m = m_{\text{tune}} + m_{\text{cert}}$, offset $\delta_0$

**Ensure:** Candidate threshold $\lambda$

1: $S_{i,j} \leftarrow s(\boldsymbol{x}_i + \boldsymbol{\epsilon}_{i,j}, y_i) \ \forall (\boldsymbol{x}_i, y_i) \in \mathcal{D}_n, \forall j \in [m_{\text{tune}}]$ for $\boldsymbol{\epsilon} \sim \psi$
2: $\lambda_{\min} \leftarrow \min_{i,j} S_{i,j}, \quad \lambda_{\max} \leftarrow \max_{i,j} S_{i,j}$
3: **while** $\lambda_{\max} - \lambda_{\min} > $ tol **do**
4: $\quad \lambda' \leftarrow (\lambda_{\min} + \lambda_{\max})/2$
5: $\quad$ **for all** $(\boldsymbol{x}_i, y_i) \in \mathcal{D}_n$ **do**
6: $\quad\quad \beta_i(\lambda') \leftarrow \frac{1}{m_{\text{tune}}} \sum_{j=1}^{m_{\text{tune}}} \mathbb{I}\big[s(\boldsymbol{x}_i + \boldsymbol{\epsilon}_{i,j}, y_i) \geq \lambda'\big]$
7: $\quad\quad \underline{\beta}_i \leftarrow \text{ClPr}\big(m_{\text{cert}} \cdot \beta_i(\lambda'), m_{\text{cert}}, 1 - \delta/|\mathcal{D}_n|\big)$
8: $\quad\quad \tilde{\beta}_i \leftarrow \text{c}^{\downarrow}[\underline{\beta}_i, \mathcal{B}]$
9: $\quad$ **end for**
10: $\quad R(\lambda') \leftarrow \frac{1}{n+1} \sum_{i=1}^n \tilde{\beta}_i$
11: $\quad$ **if** $R(\lambda') \geq 1 - \alpha + \delta + \delta_0$ **then**
12: $\quad\quad \lambda_{\max} \leftarrow \lambda'$
13: $\quad$ **else**
14: $\quad\quad \lambda_{\min} \leftarrow \lambda'$
15: $\quad$ **end if**
16: **end while**
17: $\lambda \leftarrow \lambda_{\min}$
18: **Return** $\lambda$

---

*Table 1.* Computational complexity of smoothing-based RCPs.

| Method | Calibration | Test (per point) |
|---|---|---|
| BinCP | $\mathcal{O}(|\mathcal{D}_n| \cdot n_{\text{MC}})$ | $\mathcal{O}(n_{\text{MC}})$ |
| RCP1 | $\mathcal{O}(1)$ | $\mathcal{O}(1)$ |
| FLoR$^{(1)}$ | $\mathcal{O}(|\mathcal{D}_n| \cdot n_{\text{MC}})$ | $\mathcal{O}(1)$ |

## C. Related Works and Limitations

Conformal prediction, introduced by Vovk et al. (2005), is a framework for constructing prediction sets that contain the true outcome with a user-specified probability of $1 - \alpha$. The method relies on a score function $s : \mathcal{X} \times \mathcal{Y} \mapsto \mathbb{R}$, which measures the compatibility between inputs and candidate outputs, together with an exchangeable set of held-out calibration samples. In § 2 we discuss the calibration and inference procedure, and a comprehensive tutorial can also be found in (Angelopoulos et al., 2024). Robust conformal prediction - as originally introduced by Gendler et al. (2021) - aims to extend the coverage guarantee the inputs from the same distribution perturbed with worst case noise. Besides the worst case (a.k.a. adversarial) robustness, there are orthogonal robustness frameworks proposed for CP: robustness to average noise or average adversarial examples, known as probabilistically-robust CP (Ghosh et al., 2023), and robustness to covariate (Tibshirani et al., 2019), and general distribution shifts (Barber et al., 2022). For adversarial robustness, a group of works are using Lipschitz boundedness of the networks or verifier (Jeary et al., 2024; Massena et al., 2025). These methods (i) require white-box access to the model, and (ii) they provide robustness with useful sets up to a very small radius (magnitude of perturbation). Alternatively, randomized smoothing provides black-box robustness to a much larger radii and it can apply to any model. Originally smoothing-based robustness was introduced to CP by Gendler et al. (2021), proposing to use $\hat{s}(\boldsymbol{x}, y) = \mathbb{E}_{\epsilon}[s(\boldsymbol{x} + \epsilon, y)]$ as a new score. Here the noise $\epsilon$ comes from a standard distribution, and the smooth score $\hat{s}$ changes slowly around the input. Therefore, the mean score at the clean input can be bounded by the score at the received (potentially perturbed) input. While computing $\hat{s}$ is in general intractable, we can estimate it through Monte-Carlo (MC) sampling, followed by finite sample correction. The original work did not account for finite sample error proposing a guarantee that would be only asymptotically valid. A follow-up by Yan et al. (2024) solves the issue by applying finite sample correction. Further Zargarbashi et al. (2024) proposed to use the tighter CDF-based bound and restated calibration-time robustness to derive smaller prediction sets with the same guarantee. Furthermore the authors propose robustness to label, and feature poisoning

---

**Algorithm 3** FLoR$^{(1)}$: Validating $\lambda$

---

**Require:** Calibration set $\mathcal{D}_n = \{(\boldsymbol{x}_i, y_i)\}_{i=1}^n$, score function $s : \mathcal{X} \times \mathcal{Y} \to \mathbb{R}$, smoothing distribution $\psi$, threat model $\mathcal{B}$, nominal miscoverage $\alpha$, confidence parameter $\delta$, total MC budget $m = m_{\text{tune}} + m_{\text{cert}}$, pre-computed value $\lambda$.

**Ensure:** Whether $\lambda$ results in $1 - \alpha$ robust coverage.

1: $S_{i,j}^{\text{cert}} \leftarrow s(\boldsymbol{x}_i + \boldsymbol{\epsilon}_{i,j}, y_i) \; \forall (\boldsymbol{x}_i, y_i) \in \mathcal{D}_n, \forall j \in [m_{\text{cert}}]$ for $\boldsymbol{\epsilon} \sim \psi$
2: $\beta_i(\lambda) \leftarrow \frac{1}{m_{\text{cert}}} \sum_{j=1}^{m_{\text{cert}}} \mathbb{I}[S_{i,j}^{\text{cert}} \geq \lambda] \; \forall (\boldsymbol{x}_i, y_i) \in \mathcal{D}_n$
3: $\underline{\beta}_i \leftarrow \text{ClPr}(m_{\text{cert}} \cdot \beta_i(\lambda), m_{\text{cert}}, 1 - \delta/|\mathcal{D}_n|)$
4: $\tilde{\beta}_i \leftarrow \text{c}^{\downarrow}[\underline{\beta}_i, \mathcal{B}]$
5: $R(\lambda) \leftarrow \frac{1}{n+1} \sum_{i=1}^n \tilde{\beta}_i$
6: **if** $R(\lambda) \geq 1 - \alpha + \delta$ **then**
7:     **Return** Yes
8: **else**
9:     **Return** No
10: **end if**

---

which is out of the scope of this work. All three methods are using confidence certificates (since the score function is continuous) which requires more MC samples compared to binary certificates. Zargarbashi & Bojchevski (2025) proposes a robust CP through a quantile of quantiles method (called BinCP), and show that the robustness can be attained using a single binary certificate. This allows them to reduce the needed sample-rate by an order of magnitude. By the time, BinCP is the state of the art sample-heavy robust CP.

In practice, robust CP methods must support fast inference, yet existing approaches incur substantial computational overhead at test time. To address this limitation, Zargarbashi et al. (2025) provides smoothing-based robustness that works with a single forward pass on a noise-augmented input, thereby eliminating the need for Monte Carlo estimation during inference. The resulting work is a RCP, that is equal to BinCP with modest sampling budget (e.g. in their image classification between 70 to 150 samples). While working with the same running time, RCP1 outperforms the verification-based robustness. Clearly by increasing the sampling budget in BinCP, RCP1 falls behind in usability.

**Limitations.** (i) **Small radii.** At very small perturbation radii, FLoR$^{(1)}$ can produce slightly larger sets than RCP1 due to finite-sample correction and tuning conservativeness (see Sec. 3). This is not fundamental: both effects diminish with increasing calibration sample rate, and FLoR$^{(1)}$ always outperforms RCP1 for any fixed $r > 0$ given a sufficient budget. (ii) **One-time calibration assumption.** The test-time computational advantage of FLoR$^{(1)}$ amortizes over many test points. In settings with frequent model updates or distribution shifts that require re-calibration, the extra calibration cost may not be worthwhile; RCP1 is then the recommended fallback (see § C). (iii) **Stochastic sets.** FLoR$^{(1)}$ retains the inherent stochasticity of single-sample inference; however FLoR$^{(k)}$ addresses this at the cost of higher sample rate at inference time; e.g. $k \geq 21$.

## D. Supplementary to Theory

### D.1. Proofs

**Proposition 3.1.** *Given clean calibration points* $Z_1, \ldots, Z_n$ ($Z_i = (X_i, Y_i)$), *and a potentially perturbed* $\tilde{Z}_{n+1} = (\tilde{X}_{n+1}, Y_{n+1})$, *for* $\tilde{X}_{n+1} \in \mathcal{B}(X_{n+1})$, *let* $E_i : i \in [n+1] \sim \psi$ *from a predefined smoothing scheme* $\psi$, *we have*

$$\Pr_{E_{n+1}, \mathcal{D}}[Y_{n+1} \in \mathcal{C}(\tilde{X}_{n+1})] \geq 1 - \alpha$$

*For* $\mathcal{C}(\tilde{X}_{n+1}) = \{y : s(\tilde{X}_{n+1} + E_{n+1}, y) \geq 1 - \lambda\}$ *where* $1 - \lambda$ *is defined as following: Let* $\beta_i(\lambda) = \Pr_{E_i}[s(X_i + E_i, Y_i) \geq 1 - \lambda]$ *and* $\underline{\beta}_i \leq \beta_i$ *with* $1 - \delta/|\mathcal{D}_n|$ *probability. Then*

$$\lambda = 1 - \sup\{\lambda' : \frac{1}{n+1} \sum_{i=1}^n \text{c}^{\downarrow}(\underline{\beta}_i(\lambda'), \mathcal{B}) \geq 1 - \alpha + \delta\}$$

*Proof for Proposition 3.1.* We prove through conformal risk control: As discussed, define the risk as the miscoverage

probability; i.e.

$$L(X_i; \lambda) = \max_{\tilde{X}_i \in \mathcal{B}(X_i)} Pr_{E_i}[s(\tilde{X}_i + E_i, Y_i) < 1 - \lambda]$$

which by definition is equivalent to $1 - \tilde{\beta}_i(\lambda)$. This risk upper bounded by 1, and non-increasing right continuous w.r.t. $\lambda$ as by increasing it, (as $1 - \lambda$ decreases toward 0) the probability of miscoverage decreases. It shows that the risk function complies with all conditions to be used in risk control framework.

Notably replacing $L_i(\lambda)$ with any upper bound can only increase $\lambda$, making the final expected risk even lower. From Eq. 4

$$\beta_i(\lambda) = Pr_{E_i}[s(X_i + E_i, Y_i) \geq 1 - \lambda] \quad \text{implies}$$

$$\mathrm{c}^{\downarrow}(\beta_i(\lambda), \mathcal{B}) \leq \min_{\tilde{X}_i \in \mathcal{B}(X_i)} \Pr_{E_i}[s(\tilde{X}_i + E_i, Y_i) \geq 1 - \lambda]$$

Meaning that $L(X_i; \lambda) \leq 1 - \mathrm{c}^{\downarrow}(\beta_i(\lambda), \mathcal{B})$ is a valid replacement. Through conformal risk control we have

$$\mathbb{E}[L(X_{n+1}, \lambda^*)] \leq \alpha - \delta \quad \text{for}$$

$$\lambda^* = \inf\{\lambda : \frac{1}{n+1}\big(\sum_{i=1}^{n} L(X_i; \lambda) + 1\big) \leq \alpha - \delta\}$$

Replacing the risk with a valid upper bound we have

$$\lambda_{\text{alt}} \geq \lambda \quad \text{for} \quad \lambda_{\text{alt}} =$$

$$\inf\{\lambda : \frac{1}{n+1}\big(\sum_{i=1}^{n} 1 - \mathrm{c}^{\downarrow}[\beta(\lambda), \mathcal{B}] + 1\big) \leq \alpha - \delta\}$$

$$= -\sup\{\lambda' : \frac{1}{n+1}\sum_{i=1}^{n} \mathrm{c}^{\downarrow}[\beta(\lambda'), \mathcal{B}] \geq 1 - \alpha + \delta\}$$

For the test point we have

$$\mathbb{E}[L(X_{n+1}, \lambda^{\text{alt}})] \leq \mathbb{E}[L(X_{n+1}, \lambda^*)] \leq \alpha - \delta \quad \text{i.e.}$$

$$\Pr[s(\tilde{X}_{n+1} + E_{n+1}, Y_{n+1}) \geq 1 - \lambda_{\text{alt}} | \tilde{X}_{n+1} \in \mathcal{B}(X_{n+1})] \geq 1 - \alpha + \delta$$

Estimating $\beta_i(\lambda)$ through MC sampling and computing Clopper-Pearson intervals with $1 - \delta/|\mathcal{D}_n|$ confidence result in $\delta$ failure probability of the bound through union bound. The failure probability of CP itself is $\alpha - \delta$, hence the failure of the whole setup is bounded by $(\alpha - \delta) + \delta = \alpha$. $\qquad\square$

## E. Supplementary to Experiments

**Experimental setup.** We follow Zargarbashi et al. (2025) in evaluation pipeline: For classification task we evaluate our approach on CIFAR-10, and ImageNet datasets. For CIFAR-10, we evaluate performance over 2,048 test samples, and for the ResNet and 10,000 images for the ViT models. For ImageNet, we report results for 5,000 images for ViT models and 50,000 images for ResNet models. The total number of evaluated samples does not affect the empirical conclusions. The subsets are taken exchangeably from the actual dataset. Unless specified otherwise, FLoR[(1)] uses 10,000 Monte Carlo samples during calibration, and a single MC sample during test. Same sample rate is used for BinCP (the ideal setup) both at calibration sample and test. The size of the calibration set is random between 100 to 250 and sampled exchangeably. The only effect calibration set size is on the concentration of the empirical result around the expectation – the coverage comes from a Beta distribution $\Pr[S_{n+1} \geq q \mid \mathcal{D}_n] \sim \text{Beta}((1 - \alpha) \cdot (n + 1), \alpha(n + 1)))$ and it concentrates around the expected coverage with a rate increasing by the calibration set size. The initial number of Monte Carlo samples is set to 10,000 both datasets; we subsample from these precomputed runs for other sample rates. There is one exception in Fig. 6-right where we increase the sample rate up to 60,000 for the CIFAR-10 dataset. Our results are averaged over 100 runs. In each run, 10% of the points are randomly selected as the calibration set.

**Computationally demanding setup.** For CIFAR-10, we pair a 50M-parameter diffusion model from Dhariwal & Nichol (2021) with a `ViT-B/16` model from Dosovitskiy et al. (2020), pretrained on ImageNet at $224 \times 224$ resolution and

finetuned on CIFAR-10, achieving 97.9% accuracy in the HuggingFace implementation. For ImageNet, we employ a 552M-parameter class-unconditional diffusion model followed by a BEiT-L model (305M parameters) from Bao et al. (2021), which attains 88.6% top-1 validation accuracy. Implementations are taken from the `timm` library (Wightman, 2019).

**Regression experiment.** To demonstrate that FLoR$^{(1)}$ extends naturally beyond classification, we evaluate it on the Udacity self-driving car steering dataset, using the standard absolute-residual score function and Gaussian smoothing with $\sigma = 0.15$. Table 2 reports the average prediction interval width for RCP1 and FLoR$^{(1)}$ (with $10^4$ calibration samples), confirming that FLoR$^{(1)}$ consistently produces substantially smaller intervals at non-trivial radii.

*Table 2.* Average prediction interval width (lower is better) and empirical coverage on the Udacity self-driving car regression benchmark. $\sigma = 0.15$, $10^4$ calibration samples, $\alpha = 0.1$.

| | Coverage | | Avg. Interval Width | |
|---|---|---|---|---|
| $r$ | RCP1 | FLoR$^{(1)}$ | RCP1 | FLoR$^{(1)}$ |
| 0.00 | 0.8952 | 0.9165 | 0.3567 | 0.3917 |
| 0.12 | 0.9775 | 0.9340 | 0.6772 | 0.4276 |
| 0.25 | 0.9948 | 0.9559 | 0.9220 | 0.5612 |

**Tuning sample size sensitivity.** Table 3 ablates over the tuning budget $m_{\text{tune}}$ (with fixed $m_{\text{cert}} = 10{,}000$ total samples). Increasing $m_{\text{tune}}$ beyond 500 does not consistently reduce set size and can even increase it slightly (due to wider confidence intervals in the simulated validation). The offset $\delta_0 = 0.005$ provides direct control over the failure rate; with 500 tuning samples and this default offset, zero failures are observed across all radii.

*Table 3.* Ablation over tuning budget $m_{\text{tune}}$ for FLoR$^{(1)}$. Avg. set size and failure rate (proportion of runs where the guaranteed coverage falls below $1 - \alpha$) on CIFAR-10, ResNet, $\sigma = 0.5$, $\alpha = 0.1$.

| | Avg. Set Size | | | Failure Rate | | |
|---|---|---|---|---|---|---|
| $m_{\text{tune}}$ | $r=0.12$ | $r=0.25$ | $r=0.50$ | $r=0.12$ | $r=0.25$ | $r=0.50$ |
| 50 | 3.043 | 3.424 | 4.199 | 0.04 | 0.06 | 0.84 |
| 100 | 3.053 | 3.405 | 4.193 | 0 | 0.04 | 0.70 |
| 200 | 3.056 | 3.457 | 4.194 | 0 | 0 | 0.16 |
| 500 | 3.087 | 3.402 | 4.338 | 0 | 0 | 0 |
| 1000 | 3.067 | 3.453 | 4.311 | 0 | 0 | 0 |
| 2000 | 3.042 | 3.438 | 4.373 | 0 | 0 | 0 |
| 5000 | 3.170 | 3.552 | 4.581 | 0 | 0 | 0 |

**Supplementary reports.** We report the numerical value of the result for ImageNet dataset in Table 4 (ResNet model) and Table 5 (diffusion + ViT model). In the same order we report the results for CIFAR-10 dataset in Table 6, and Table 7.

**FLoR$^{(k)}$.** Recall that FLoR$^{(k)}$ explicitly calibrates to account for the downstream majority-vote aggregation, by enforcing a margin-above-half constraint on the smoothed coverage probabilities. This effectively shifts calibration from controlling single-sample coverage (which calibrates with the objective over expected $\beta_{n+1}$) to directly enabling majority vote success (ensuring that in $1 - \alpha$ cases $\beta_{n+1}$ remains above the margin). Fig. 7 empirically examines how this mechanism interacts with the aggregation rate $k$ and the offset parameter $\alpha_0$ across three representative robustness radii. Formally for a fixed value of $\alpha_0$, and $k$, the value $\beta_0$ has a closed form of:

$$\beta_0 = \inf\left\{ p \in [1/2, 1] \ : \ (1 - \alpha + \alpha_0) \cdot \left(1 - \Phi_{\text{bin}}\left(\frac{k-1}{2}, k, \frac{1}{2} + p\right)\right) \ > \ 1 - \alpha \right\}.$$

Picking any value above this closed form only increases the set size.

The top row reports relative improvements over the single-sample baseline FLoR$^{(1)}$. We observe a broad region in which FLoR$^{(k)}$ consistently yields smaller prediction sets, with the largest gains attained for moderate test-time sample rates and small but nonzero offsets $\alpha_0$. This behavior reflects the intended role of $\alpha_0$: introducing a mild safety margin over $1 - \alpha$ is necessary as the majority vote has a probability which by definition is lower than one. Increasing $\alpha_0$ clearly decreases the needed $\beta_0$ but enforces larger set size through calibrating over a higher coverage. What we observe is that it is better not to increase $\alpha_0$ more than a very small safety margin. As $k$ increases, the improvement region becomes more structured and gradually saturates, indicating diminishing returns once the variance of the majority-vote estimator is sufficiently reduced.

*Table 4.* Average prediction set size (mean $\pm$ std) as a function of perturbation radius $r$. The results are for the ImageNet dataset, ResNet model

| $r$ | BinCP | RCP1 | FLoR$^{(1)}$ |
|------|-------|------|--------------|
| 0.00 | $10.770 \pm 7.584$ | $8.605 \pm 5.110$ | $12.451 \pm 8.123$ |
| 0.06 | $11.243 \pm 7.912$ | $10.396 \pm 6.896$ | $13.128 \pm 8.597$ |
| 0.12 | $11.732 \pm 8.240$ | $19.595 \pm 14.091$ | $13.786 \pm 8.960$ |
| 0.18 | $12.265 \pm 8.592$ | $23.810 \pm 18.245$ | $14.673 \pm 9.195$ |
| 0.25 | $12.911 \pm 9.041$ | $45.019 \pm 28.125$ | $15.905 \pm 9.931$ |
| 0.37 | $14.133 \pm 9.869$ | $69.886 \pm 50.844$ | $18.693 \pm 12.025$ |
| 0.50 | $15.570 \pm 10.795$ | $144.226 \pm 105.848$ | $24.076 \pm 16.621$ |

*Table 5.* Average prediction set size (mean $\pm$ std) as a function of perturbation radius $r$. The results are for the ImageNet dataset, Diffusion + ViT model

| $r$ | BinCP | RCP1 | FLoR$^{(1)}$ |
|------|-------|------|--------------|
| 0.00 | $3.658 \pm 0.624$ | $3.989 \pm 0.990$ | $6.084 \pm 1.912$ |
| 0.06 | $3.877 \pm 0.685$ | $6.464 \pm 2.093$ | $6.735 \pm 2.198$ |
| 0.12 | $4.088 \pm 0.738$ | $8.414 \pm 2.390$ | $7.507 \pm 2.535$ |
| 0.18 | $4.345 \pm 0.802$ | $10.347 \pm 2.341$ | $8.401 \pm 2.833$ |
| 0.25 | $4.672 \pm 0.899$ | $26.859 \pm 15.663$ | $9.802 \pm 3.335$ |
| 0.37 | $5.383 \pm 1.099$ | $79.228 \pm 108.014$ | $13.367 \pm 4.348$ |
| 0.50 | $6.304 \pm 1.361$ | $197.633 \pm 150.188$ | $20.765 \pm 7.514$ |

*Table 6.* Average prediction set size (mean $\pm$ std) as a function of perturbation radius $r$. Results for CIFAR-10 dataset and ResNet model.

| $r$ | BinCP | RCP1 | FLoR$^{(1)}$ |
|------|-------|------|--------------|
| 0.00 | $2.143 \pm 0.187$ | $2.470 \pm 0.257$ | $2.680 \pm 0.165$ |
| 0.06 | $2.238 \pm 0.212$ | $2.717 \pm 0.269$ | $2.791 \pm 0.198$ |
| 0.12 | $2.336 \pm 0.201$ | $2.912 \pm 0.301$ | $2.921 \pm 0.180$ |
| 0.18 | $2.475 \pm 0.185$ | $3.228 \pm 0.328$ | $3.097 \pm 0.162$ |
| 0.25 | $2.642 \pm 0.207$ | $3.743 \pm 0.578$ | $3.291 \pm 0.176$ |
| 0.37 | $2.889 \pm 0.207$ | $4.537 \pm 0.583$ | $3.647 \pm 0.192$ |
| 0.50 | $3.266 \pm 0.220$ | $5.413 \pm 0.754$ | $4.165 \pm 0.214$ |

*Table 7.* Average prediction set size (mean $\pm$ std) as a function of perturbation radius $r$. Results for CIFAR-10 dataset and Diffusion + ViT model.

| $r$ | BinCP | RCP1 | FLoR$^{(1)}$ |
|------|-------|------|--------------|
| 0.00 | $2.335 \pm 0.652$ | $3.166 \pm 0.483$ | $3.319 \pm 0.448$ |
| 0.06 | $2.507 \pm 0.690$ | $3.877 \pm 0.314$ | $3.683 \pm 0.495$ |
| 0.12 | $2.716 \pm 0.750$ | $4.284 \pm 0.245$ | $4.054 \pm 0.544$ |
| 0.18 | $2.939 \pm 0.789$ | $4.657 \pm 0.267$ | $4.471 \pm 0.598$ |
| 0.25 | $3.193 \pm 0.828$ | $6.856 \pm 1.758$ | $4.944 \pm 0.598$ |
| 0.37 | $3.667 \pm 0.861$ | $8.327 \pm 1.060$ | $5.729 \pm 0.532$ |
| 0.50 | $4.223 \pm 0.911$ | $8.935 \pm 0.825$ | $6.453 \pm 0.556$ |

Fig. 8 visualizes stochasticity, in form of *input-conditional* set size probability. To compute this probability, we first sample the calibration set, and calibrate each method (notably for BinCP, FLoR$^{(1)}$, and FLoR$^{(k)}$ we use $10^4$ samples per each point), then for 10 different runs, we run each method with its predefined test-time sample rate, which applies to BinCP, and FLoR$^{(k)}$, as for RCP1, and FLoR$^{(1)}$ the test-time sample rate is always 1. We compute the probability the frequency of the discrete events (set size values). Although this is a MC-sampling estimation with very low sample, we do not need the probabilities to be precise as we only want to compare the stochasticity of each method. Each column corresponds to a fixed test input $\boldsymbol{x}_{n+1}$, and the intensity profile along the $y$-axis depicts the distribution of $|\mathcal{C}_r(\boldsymbol{x}_{n+1})|$ induced solely by the sampling randomness (with calibration performed using $10^4$ samples). As expected, both RCP1 and FLoR$^{(1)}$, stochasticity in the prediction sets while BinCP remains more deterministic for all sample rates. Interestingly in all empirical evaluations, FLoR$^{(k)}$ is more deterministic compared to BinCP. Notably by increasing the test-time sample rate (from SR$=$ 21 to SR$=$ 101) progressively BinCP and FLoR$^{(k)}$ sharpen their conditional distribution and reduces both the frequency of large outliers and the column-wise dispersion.

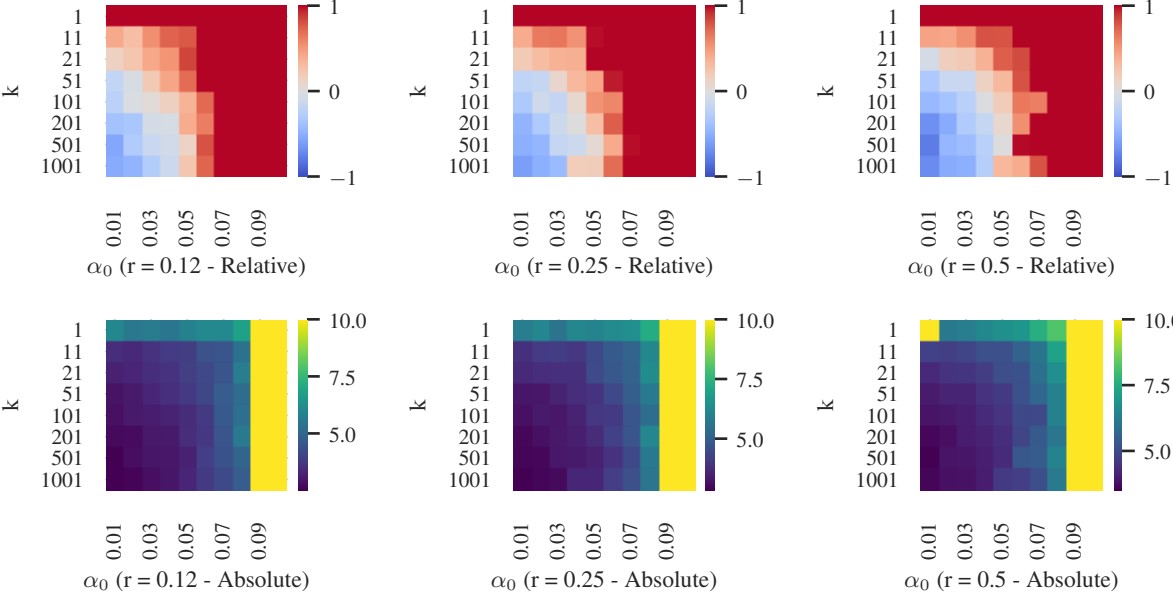

*Figure 7.* Ablation study on hyperparameters $\alpha_0$ and $k$ for FLoR$^{(k)}$: We evaluate the average set size of FLoR$^{(k)}$ across different values of the initial coverage offset $\alpha_0$ and the test-time sample rate $k$ at three representative radii ($r \in \{0.12, 0.25, 0.5\}$). [Top row] Relative improvement over FLoR$^{(1)}$ (negative values indicate smaller sets - note that the colorbar is clipped at -1 to 1 meaning that areas of same extreme color might project difference more than 1, or less than -1). [Bottom row] Absolute average set sizes. All results are for CIFAR-10 with $\sigma = 0.5$ and ResNet model.

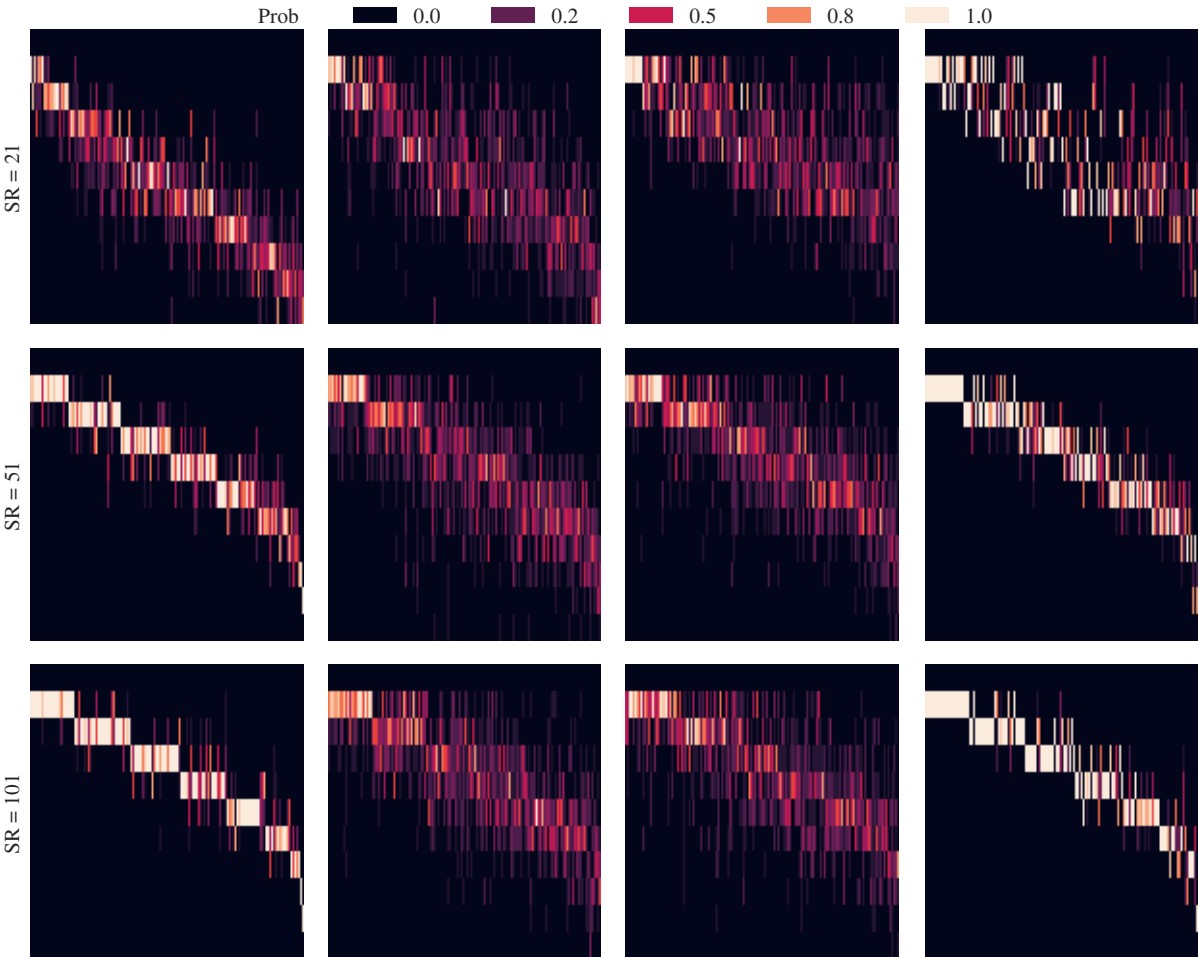

*Figure 8.* The distribution of prediction set sizes, conditional to test inputs. Each column of the plot refers to a single $\boldsymbol{x}_{n+1}$, and the $y$-axis shows the set size. The brightness of each cell $i, j$ shows the probability $|\mathcal{C}(\boldsymbol{x}_{n+i})| = j$. The methods are [from left to right] BinCP, RCP1, FLoR$^{(1)}$, and FLoR$^{(k)}$. The results are for CIFAR-10 dataset, ResNet model with $\sigma = 0.5$, and $r = 0.12$, with $10^4$ calibration time samples. The test time sample rate is shown in the beginning of each row.

