# OpenReview forum: "Front-Loaded Robust Conformal Prediction: Heavy Calibration, Minimal Test-Time Cost"
_ICML.cc/2026/Conference — ICML 2026 regular_

### Official Review · Reviewer_CQ1p · 2026-02-15

**Soundness:** 3
**Presentation:** 2
**Significance:** 2
**Originality:** 2
**Overall Recommendation:** 4
**Confidence:** 3

**Summary:**

The paper proposes Front-Loaded Robust Conformal Prediction (FLoR) for randomized-smoothing Robust Conformal Prediction (RCP). Prior methods either use heavy Monte Carlo sampling at calibration and test (small sets but slow inference) or single-sample certification (fast but conservative, larger sets). FLoR shifts computation to the one-time calibration stage by using more sampling there while keeping test-time sampling minimal, yielding smaller robust prediction sets at the same inference cost. It also introduces FLoR(k), a low-sample majority-vote variant to reduce randomness in set size. Empirically, FLoR produces smaller sets than prior fast baselines under matched budgets, especially at larger robustness radii.

**Compliance With Llm Reviewing Policy:**

Affirmed.

**Final Justification:**

The authors' rebuttal addressed all my concerns.

**Key Questions For Authors:**

1. see weaknesses
2. Please provide more complete credit and contextualization within prior work. For example, in the first two paragraphs of the introduction, key statements about conformal prediction and robust conformal prediction are made with no citations.
3. The proposed “certification + (smaller) tuning budget” pipeline introduces additional hyperparameter choices. Please provide more detailed hyperparameter sensitivity analysis to strengthen the results.
4. In the small-noise/small-radius setting, FLoR sometimes yields slightly larger prediction sets than RCP1. Could the authors clarify the intuition why larger radius is better for FloR and if there are any theoretical insights here?
5. Since the method appears applicable beyond classification, it would strengthen the paper to include at least one regression experiment.

**Limitations:**

yes

**Strengths And Weaknesses:**

Strengths

1. The paper tackles an important and timely problem: robust uncertainty quantification under noise/adversarial perturbations.
2. The theoretical guarantees are clearly stated and appear sound, and the empirical results are thorough and convincing.
3. The figures are easy to follow and the visualizations effectively communicate the key takeaways.

Weaknesses
1.  The main advantage relies on settings where calibration is a one-time procedure and the number of test points is large enough to amortize calibration cost; in scenarios with frequent model updates, distribution shift, or repeated re-calibration, the benefit may diminish. More discussion of how this framework relates to (and could be integrated with) broader ML robustness/calibration literature would strengthen the paper.
2.  Although the method shifts computation to calibration, it still relies on substantial Monte Carlo sampling to achieve smaller sets. When budgets are tight, the resulting procedure may become less effective or more conservative.

---

> ### Author Rebuttal · Authors · 2026-03-30
>
> We thank the reviewer for the reading and constructive feedback. Below we address each point in turn.
>
> **W1.** The point about settings with frequent re-calibration (e.g., model updates or inductive graph node classification) is valid. However, we make two points:
>
> 1. *Other baselines share the same limitation.* BinCP (the current SOTA) requires heavy MC sampling at *both* calibration and test time, so its overhead also recurs in re-calibration. RCP1 works with single forward pass, but its larger set size coming from Jensen's inequality is irreducible regardless of the budget. The same compute-vs-set-size trade-off is therefore present across all existing methods in frequent re-calibration settings.
>
> 2. *Still in many such settings, re-calibration is still infrequent enough to amortize.* For example, in streaming deployment with periodic model updates, one can choose FLoR(1) when test throughput is high, and fall back to RCP1 o.w. Still this is an interesting orthogonal question to ask "how to achieve efficient re-calibration using the same statistics and keeping local robustness guarantees".
>
> We will elaborate the connection of RCP with calibration / robustness in a more detailed way within related works in camera ready.
>
> **W2.** We agree. The same limitation however applies to all other baselines. The paper's core contribution is to improve the efficiency-vs-set-size trade-off by concentrating cost at calibration. The purpose of FLoR(1) and FLoR(k) is to offer a better trade-off.
>
> **Q2.** Thank you for pointing this out. We will add the missing citations in the camera ready.
>
> **Q3.** Analogously to randomized smoothing, where a small sample is used to guide for the top class before the full certification step, our tuning set allows to simulate the binary search over $\lambda$ in Prop. 3.1 in a cheaper way. Together with the conservativeness offset $\delta_0$ (set to 0.5% by default), this allows us to avoid repeated re-sampling while ensuring that a single run of Prop. 3.1 suffices.
>
> We ran an ablation over the tuning set size, using total of 10,000 samples per calibration point. "Failure" refers to cases where the constraint in Prop. 3.1 is not satisfied at the target coverage (e.g., yields 89% instead of 90%), requiring the process to be repeated. As the table shows, increasing the tuning set size does not consistently reduce set size; large values can even increase it due to wider confidence intervals. The offset $\delta_0$ provides direct control over the failure rate.
>
> | Tuning size | Avg Size: r=0.12 | r=0.25 | r=0.50 | Fail: r=0.12 | r=0.25 | r=0.50 |
> |:--|--:|---:|---:|---:|---:|---:|
> | 50    | 3.043 | 3.424 | 4.199 | 0.04 | 0.06 | 0.84 |
> | 100   | 3.053 | 3.405 | 4.193 | 0    | 0.04 | 0.70 |
> | 200   | 3.056 | 3.457 | 4.194 | 0    | 0    | 0.16 |
> | 500   | 3.087 | 3.402 | 4.338 | 0    | 0    | 0    |
> | 1000  | 3.067 | 3.453 | 4.311 | 0    | 0    | 0    |
> | 2000  | 3.042 | 3.438 | 4.373 | 0    | 0    | 0    |
> | 5000  | 3.170 | 3.552 | 4.581 | 0    | 0    | 0    |
>
> **Q4.**  Both methods use the same definition for prediction set; they differ only in how $1 - \lambda$ is computed. The gap is governed by three factors:
>
> 1. **Jensen's gap:** FLoR(1) directly estimates $E[c^\downarrow(\beta)]$ via Monte Carlo, while RCP1 uses $c^\downarrow(E[\beta])$. By Jensen's inequality (since $c^\downarrow$ is convex), $E[c^\downarrow(\beta)] \geq c^\downarrow(E[\beta])$, so FLoR(1) obtains a strictly tighter threshold and thus smaller prediction sets. Via Taylor expansion: $E[c^\downarrow(\beta)] \approx c^\downarrow(E[\beta]) + \tfrac{1}{2} c^{\downarrow''}(E[\beta]) \operatorname{Var}[\beta]$. The second derivative of the Gaussian certified lower bound is 0 at $r = 0$, increases to a maximum at $r^\star = \frac{\Phi^{-1}(1-\alpha) + \sqrt{(\Phi^{-1}(1-\alpha))^2 + 4}}{2}$, and returns to 0 as $r \to \infty$, but is strictly positive for all intermediate $r$.
>
> 2. **Finite-sample correction:**  higher target coverage + Clopper–Pearson confidence bound, both inflating the set size. This effect diminishes with increasing sample rate.
>
> 3. **Tuning conservativeness:** Conservative implementation to ensure a single run of Prop. 3.1 suffices.
>
> Factors (2) and (3) both inflate the set size relative to the asymptotic ideal. At small $r$, the Jensen gap (1) is near zero, so (2) and (3) dominate — RCP1 outperforms FLoR(1). At larger $r$, the Jensen gap grows and FLoR(1) gains the advantage. The effect of (2) diminishes with increased sample rate, though the required sample rate grows quickly for small radii.
>
> **Question 5.** For regression, we compare RCP1 and FLoR(1) on the Udacity self-driving car dataset with 50,000 calibration samples and the standard residual score function. $\sigma = 0.15$.
>
> | $r$ | Coverage: RCP1 | FLoR(1) | Avg Set Size: RCP1 | FLoR(1) |
> |---:|---:|---:|---:|---:|
> | 0    | 0.8952 | 0.9165 | 0.3567 | 0.3917 |
> | 0.12 | 0.9775 | 0.9340 | 0.6772 | 0.4276 |
> | 0.25 | 0.9948 | 0.9559 | 0.9220 | 0.5612 |

---

> > ### Author Rebuttal · Reviewer_CQ1p · 2026-04-02
> >
> > Thanks the authors for the detailed response. I'll raise my score.

---

### Official Review · Reviewer_UUYf · 2026-03-02

**Soundness:** 2
**Presentation:** 1
**Significance:** 2
**Originality:** 3
**Overall Recommendation:** 4
**Confidence:** 2

**Summary:**

The authors address the problem of computing Conformal Prediction intervals under adversarial attacks. Existing methods use Monte Carlo sampling to smooth the score function and evaluate worst-case scenarios, and are computationally expensive at both calibration and test time. The proposed approach, which leverages Conformal Risk Control to bound the test-time coverage probability, is only expensive at calibration time.

**Compliance With Llm Reviewing Policy:**

Affirmed.

**Final Justification:**

The authors' rebuttal addressed all my concerns. I have raised my score to 4.

**Key Questions For Authors:**

- Are there assumptions about the data-generating distribution in RCP (such as Lipschitz continuity)?
- Does the claim that CP validity breaks *severely under even a very small magnitude of perturbation in the input space* include the reweighting technique for covariate shift [1]? Or is RCP also robust to general shifts where both $P(X)$ and $P(Y|X)$ change?
- Are the radii in the introduction the noise perturbation radii? What happens if the noise perturbation set is not a ball?
- What does *them* in [2] refer to?
- How do you guarantee that the conformalized worst-case expected coverage can be transferred to test samples? Does this require exchangeability between MC samples and test points?
- Is there an intuitive reason why the majority voting does not break the *theoretical guarantees* of one-sample smoothing? Is this related to the smoothing distribution of Proposition 4.1? How does such a smoothing distribution affect the bounds?
- What is the * certified lower bound function * mentioned in the review of RCP1?

[1]
Tibshirani, R. J., Foygel Barber, R., Candes, E., and Ramdas, A. Conformal prediction under covariate shift. Advances in neural information processing systems, 32, 2019.

[2]
*we estimate them from calibration points through Monte Carlo (MC) sampling.*

**Limitations:**

Yes

**Strengths And Weaknesses:**

**Strengths**
- Studying the behaviour of CP bounds under controlled probabilistic perturbation is an interesting idea.
- Assuming a higher computational budget in the calibration stage is a realistic setup. Exploiting it to augment the calibration set by MC sampling is a powerful general approach, even outside the adversarial bounded-norm framework.


**Weaknesses**
- The material could have been presented more linearly. The abstract and the introduction contain possibly ambiguous concepts such as *miscalibration*, *worst-case noise*, or *robustness at larger radii*. Some sentences, e.g. [1],  are hard to interpret.
- The authors should mention earlier the type of assumptions made by RCP. For example, they should say earlier that the noise is assumed to have a bounded norm.
- A more detailed explanation of the certified probability bounds mentioned in the introduction would be helpful for readers unused to the adversarial setup.
- The explanation of randomized smoothing in Section 2 is hard to follow.
- It is unclear under which assumptions the proposed guarantees hold. A comparison with the requirements of other methods would be helpful.

[1]
*The coverage probability in calibration points can represent the test points.*

---

> ### Author Rebuttal · Authors · 2026-03-30
>
> We thank the reviewer for the feedback and commit to incorporating the following clarifications in the camera-ready version.
>
> **W1, W2, and W3.** We appreciate your feedback. Bounded-norm noise is a standard threat model in the robustness literature, since unbounded perturbations can arbitrarily transform one image into another. That said, we agree that the imperceptibility motivation for bounded noise should be introduced earlier. We will also add an intuitive explanation of the certified bound early in the introduction.
>
> **W4 and W5.** We kept the discussion of randomized smoothing brief, as it is already well-covered in prior works. We will add a high-level introduction to the camera ready but for now, we refer the reviewer to Section C of [3] for a detailed intuitive account. Regarding assumptions: There are only two assumptions — (1) exchangeability of the clean inputs $x_1, \ldots, x_{n+1}$, standard assumption in conformal prediction; and (2) $\tilde{x}_ {n+1} \in \mathcal{B}(x_ {n+1})$ which is the standard adversarial robustness threat model also adopted by all baseline methods (see Section 2). There are no additional assumptions on the model, score function, or data-generating distribution.
>
> **Q1.** There are no assumptions on the data-generating distribution — CP is distribution-free. Regarding Lipschitz continuity: the smoothed model is Lipschitz continuous by construction (a consequence of smoothing), regardless of the underlying base model. FLoR and all baseline methods are both distribution-free and model-agnostic.
>
> **Q2.** There is an important distinction between covariate shift and local adversarial perturbation. Under covariate shift, the model receives clean inputs but the distribution $P_X$ changes. Under adversarial perturbation, $P_X$ and $P_{Y|X}$ remain unchanged, but we receive a perturbed version of the clean input, bounded by the threat model. Addressing each part of the question:
>
> > Does the claim that CP validity breaks severely under even a very small magnitude of perturbation include the reweighting technique for covariate shift [1]?
>
> Yes, it still breaks. Covariate-shift reweighting (e.g., weighted quantile) adjusts the threshold for marginal shifts in $P_X$, but adversarial perturbation exploits the instability of the conformity score at the per-input level — it can sharply reduce the conformal p-value for the true label even when the marginal distribution is unchanged (see Figure 1-left of [4]).
>
> > Is RCP also robust to general shifts where both $P_X$ and $P_{Y|X}$ change?
>
> In practice, yes — empirical coverage typically increases above the nominal level $1-\alpha$, providing a buffer. However, a direct theoretical connection between local adversarial robustness and global distributional shift is an orthogonal and challenging problem; [7] offers a partial answer.
>
> **Q3.** Yes, the radii refer to the perturbation ball radius ($\ell_2$). Following the recipes in [5, 6], the framework extends to other perturbation sets and smoothing distributions beyond $\ell_2$ balls.
>
> **Q4.** "Them" refers to the coverage probabilities of future test points.
>
> **Q5.** The guarantee follows from two steps: (1) exchangeability — random augmentation preserves exchangeability; and (2) a certified lower bound on the probability of covering the perturbed test point, derived from randomized smoothing. Both are model- and function-agnostic.
>
> **Q6.** Majority voting would break the one-sample guarantee if applied post hoc. The key in Prop. 4.1 is that calibration is made *aware of the majority vote*: instead of enforcing $E[\tilde{\beta}_ {n+1}] \geq 1-\alpha$, the method enforces a stronger condition — that on average $1-\alpha$ true labels pass a majority vote — targeting a per-point probability of at least $1 - \Phi_{\mathrm{bin}}\left(\tfrac{k-1}{2},\, k,\, \tfrac{1}{2} + \beta_0\right)$.
>
> The smoothing distribution $\psi$ affects the certified probabilities and thus the certified margin size, but the majority-vote validity itself comes from the binomial aggregation argument and does not depend on the particular choice of $\psi$.
>
> **Q7.** We refer to the second paragraph of the second column on page 3 (before the *Bounds from randomized smoothing* paragraph) for the formal definition. A certified lower bound function $c^\downarrow$ takes a smooth probability as input and returns a value guaranteed to lower-bound it at *any* point within the perturbation ball. Randomized smoothing provide such functions.
>
> [3] Zargarbashi et al (2024). Robust yet efficient conformal prediction sets.
>
> [4] Zargarbashi, et al (2025). One sample is enough to make conformal prediction robust.
>
> [5] Yang et al. (2020). Randomized smoothing of all shapes and sizes.
>
> [6] Lee et al (2019). Tight certificates of adversarial robustness for randomly smoothed classifiers.
>
> [7] Aolaritei et al. (2025). Conformal prediction under Lévy–Prokhorov distribution shifts: Robustness to local and global perturbations.

---

> > ### Author Rebuttal · Reviewer_UUYf · 2026-04-02
> >
> > Many thanks. I will raise my score to 4.

---

### Official Review · Reviewer_UEhx · 2026-03-11

**Soundness:** 2
**Presentation:** 1
**Significance:** 3
**Originality:** 3
**Overall Recommendation:** 4
**Confidence:** 3

**Summary:**

The paper proposes an approach to achieve the trade-off between test-time efficiency and prediction set size while deploying current RCP methods. The proposal is to spend computationally heavy estimation during calibration while keeping test-time sampling extremely small. This way, paper tries to resolve practical problem of deploying current RCP approaches in practice.

Conceptually, the paper exploits the ideas of RCP1 (Zargarbashi et al., 2025) based on Jensen's inequality and argues that RCP1 produces unnecessarily conservative prediction sets leading to huge sizes. Then, authors proposes that we can avoid this by estimating bounds using the ideas from conformal risk control. Two version are proposed leveraging this idea.

Experiments are provided to support the proposed methods and empirical advantage of reduced prediction set sizes with minimal test time sampling are demonstrated.

**Compliance With Llm Reviewing Policy:**

Affirmed.

**Final Justification:**

Authors satisfactorily addressed the concerns. In my opinion, paper provides novel idea to challenging problem of deploying RCP and will be of interest to the readers of ICML. Based on the author's response, I have raised my score.

**Key Questions For Authors:**

1. The method introduces additional probabilistic constraints which might loose the guarantees. It is not clear to me why proposed approach are guaranteed to lead to smaller set sizes for large radii. It is also counterintuitive that proposed approaches do not work for small radii (small perturbations).

2. Does the paper consider computational cost of tuning additional parameters during the comparisons?

3. On Lines 246-247 (Right column), it says "In case the validation fails to ensure that, we have to repeat the process once again". This issue again highlights weakness 3 above, as tuning additional parameters is heuristically presented.  Can the procedure be made rigorous further?

**Limitations:**

No, the paper could discuss the limitations of both approaches, especially when they can fail (paper mentions small radii), and also effect of tuning of hyperparameters.

**Strengths And Weaknesses:**

**Strengths**

1. Experiments demonstrate the benefits of the proposed approaches.

2. The paper is providing a solution to a practical challenge of deploying recent RCP methods.

---

**Weaknesses**

1. The paper presentation could be improved the ideas more clearly, avoiding being mathematically dense notations. I do not understand the idea of how conformal risk control works as paper does not explain what is it trying to achieve, might be due to presentation issue. Some notations are introduced earlier while reference come later in the text (for instance, "RCP1" on line 078).

2. The set size reduction claim is purely empirical, and heuristically justified via looseness of Jensen's inequality. But, the method introduces additional probabilistic constraints which might loose the guarantees. It is not clear to me why proposed approach are guaranteed to lead to smaller set sizes for large radii.

3. The method in order to avoid sampling at test time introduces additional parameters to tune.

---

> ### Author Rebuttal · Authors · 2026-03-30
>
> We thank the reviewer for the comments. We address each point below and commit to incorporating these clarifications in the camera-ready version.
>
> **W1.** We appreciate the feedback on readability and will improve the presentation in the camera ready. Briefly, Conformal Risk Control (CRC) [1] generalizes split conformal prediction beyond miscoverage: if the loss is monotone non-increasing in a parameter $\lambda$ and the data are exchangeable, then the threshold $\hat{\lambda}$ selected via Eq. 2 guarantees that the expected future risk does not exceed the pre-specified level $\alpha$. This is the recipe we use in place of the standard quantile step.
>
> By "RCP1" we refer to the method in [2]. In the camera ready, we will ensure all methods are introduced with their corresponding citation at first occurrence.
>
> **W2.** There are no additional constraints in the theory or implementation beyond what is described in Prop. 3.1. The following explains why FLoR(1) produces smaller sets than RCP1 at large radii.
>
> Both methods use the same definition prediction set; they differ only in how $1 - \lambda$ is computed. Three factors govern the comparison:
>
> 1. **Jensen's gap:** FLoR(1) estimates $E[c^\downarrow(\beta)]$ directly via Monte Carlo, while RCP1 uses $c^\downarrow(E[\beta])$. Since $c^\downarrow$ is convex over the relevant range, Jensen's inequality gives $E[c^\downarrow(\beta)] \geq c^\downarrow(E[\beta])$, so FLoR(1) obtains a strictly tighter threshold, yielding smaller prediction sets. Via Taylor expansion: $E[c^\downarrow(\beta)] \approx c^\downarrow(E[\beta]) + \tfrac{1}{2} c^{\downarrow''}(E[\beta]) \operatorname{Var}[\beta]$. The second derivative of the Gaussian certified lower bound is 0 at $r = 0$, peaks at $r^\star = \frac{\Phi^{-1}(1-\alpha) + \sqrt{(\Phi^{-1}(1-\alpha))^2 + 4}}{2}$, and returns to 0 as $r \to \infty$, but is strictly positive for all intermediate $r$.
>
> 2. **Finite-sample correction:** FLoR(1) calibrates to a higher target coverage and applies a Clopper–Pearson lower confidence bound, both of which introduce additional conservativeness.
>
> 3. **Tuning conservativeness:** The tuning step is designed to ensure a single certification run via Prop. 3.1 suffices, which further increases the conservativeness of the selected $\lambda$.
>
> Factors (2) and (3) both inflate the set size relative to the asymptotic ideal. At small $r$, the Jensen gap (1) is negligible, so (2) and (3) dominate — this is why RCP1 outperforms FLoR(1) at small radii. At larger $r$, the Jensen gap grows and FLoR(1) recovers the advantage. The underperformance at small radii is therefore not a fundamental limitation of the method but a consequence of the finite-sample implementation, which diminishes with increasing sample rate.
>
> **W3.** The tuning step serves as a cheap proxy for the binary search over $\lambda$ that Prop. 3.1 already requires. The additional h-params are the price we pay for increased efficiency compared to binary search. This is similar to the standard practice in randomized smoothing, where a small number of samples are used to identify a candidate parameter before the full certification step. Here, the tuning batch identifies a candidate $\lambda$, which is then validated exactly once against the constraint in Prop. 3.1. Because any $\lambda$ satisfying that constraint carries the full statistical guarantee — the supremum is only to find the most efficient such $\lambda$ — the guarantee depends entirely on the final validation step, not on how the candidate was found. In practice, the tuning step took under 500 milliseconds across all CIFAR-10 runs. We do not need to tune the for any specific additional parameters. We set $\delta_0=0.005$ and tuning samples to 500 which empirically results in no errors, and no need to re-run the process.
>
> **Q1.** We hope our responses to W2 and W3 address this question. If there is a specific probabilistic constraint that remains unclear, please let us know and we will elaborate further.
>
> **Q2.** The tuning step runs with the same complexity to Prop. 3.1 but with a much smaller sample budget, significantly reducing computational cost. Across all CIFAR-10 runs, the wall-clock time for tuning was under 500 milliseconds.
>
> **Q3.** We ran an ablation over the tuning set size. With 500 tuning samples and the default offset $\delta_0 = 0.005$, all tests pass in a single calibration step. Also to stay more on the safe side, one can slightly increasing $\delta_0$. Even when the constraint in Prop. 3.1 is not met at the target coverage, the outcome is very close (e.g., the guaranteed probability is 89% instead of 90%), and the process simply repeats once if that guarantee is not acceptable.
>
> [1] Angelopoulos, A. N., Bates, S., Fisch, A., Lei, L., & Schuster, T. (2024). Conformal risk control. ICLR 2024.
>
> [2] Zargarbashi, S. H., Akhondzadeh, M. S., & Bojchevski, A. (2025). One sample is enough to make conformal prediction robust. NeurIPS 2025.

---

> > ### Author Rebuttal · Reviewer_UEhx · 2026-04-02
> >
> > I thank the authors for their detailed response. I will raise the score. I have no further questions.

---

### Official Review · Reviewer_exuQ · 2026-03-13

**Soundness:** 3
**Presentation:** 3
**Significance:** 3
**Originality:** 3
**Overall Recommendation:** 4
**Confidence:** 3

**Summary:**

This paper addresses the computational efficiency and set-size trade-off in Randomized Smoothing-based Robust Conformal Prediction (RCP). Existing methods present a strict dichotomy: BinCP yields small prediction sets but requires computationally expensive Monte Carlo (MC) sampling at both calibration and test time, whereas RCP1 uses a single test sample but produces overly conservative prediction sets due to its reliance on Jensen's inequality.
The authors propose a "Front-Loaded" approach, FLoR(1), which shifts the computational burden entirely to the one-time calibration phase. By leveraging Conformal Risk Control (CRC) and heavy MC sampling during calibration, FLoR(1) directly estimates the worst-case expected coverage. This circumvents the loose bounds of RCP1, reducing the average set size while maintaining a minimal single-sample test-time cost. Furthermore, to mitigate the inherent stochasticity of single-sample prediction sets, the authors introduce FLoR(k). FLoR(k) integrates a majority-voting scheme into the calibration objective, providing valid coverage guarantees under a low test-time sampling budget. Empirical evaluations on CIFAR-10 and ImageNet confirm that FLoR(1) and FLoR(k) achieve smaller set sizes than RCP1, and offer better efficiency than BinCP at restricted test-time budgets.

**Compliance With Llm Reviewing Policy:**

Affirmed.

**Final Justification:**

Thank the authors for the detailed responses. I would like to keep my score unchanged. I have read the other reviewers’ comments and will keep track of the discussion between the authors and other reviewers.

**Key Questions For Authors:**

See above

**Limitations:**

See above

**Strengths And Weaknesses:**

Strength
1. The paper targets a very natural gap in the current smoothing-RCP literature: RCP1 is cheap at inference but conservative, whereas BinCP can be tighter but expensive at test time. The core premise—that calibration is a one-time pre-processing step and can thus absorb higher computational overhead to accelerate test-time inference—is highly pragmatic for real-world model deployment.
2. The authors correctly identify stochastic prediction sets as a real issue in single-sample methods, and they do not merely add voting heuristically; instead they recalibrate for majority vote. That is a meaningful methodological extension, and the empirical plots on set-size variability make the problem concrete.
3. Across ImageNet/CIFAR-10 and different backbone setups, the paper consistently reports smaller sets for FLoR(1) than RCP1, especially at larger radii, and shows that increasing calibration sample rate improves FLoR(1), which is exactly the claimed mechanism.

Weakness
1. The practical procedure for selecting \lambda is considerably more involved than the main theory suggests.
 While the core proposition is clean, the actual implementation relies on a nontrivial tuning/validation pipeline with separate budgets (m_{\text{tune}}, m_{\text{cert}}), Clopper–Pearson lower bounds, an additional offset \delta_0, and potentially repeated validation if the candidate threshold fails. This makes the method substantially more cumbersome in practice, and it is not fully clear that the paper provides a comparably clean end-to-end theoretical account for the complete threshold-selection procedure actually used in experiments. In addition, this tuning process appears computationally expensive in its own right.
2. FLoR(k) appears quite sensitive to hyperparameter choices, and the required tuning may be expensive.
 The method introduces multiple coupled hyperparameters, notably k, \alpha_0, and \beta_0, with the calibration condition explicitly depending on their interaction. The ablation in Figure 7 suggests that performance can vary substantially across choices of k and \alpha_0, which raises concerns about robustness and ease of use in practice. Since identifying good settings would likely require a grid search over these parameters, the practical computational burden of FLoR(k) may be higher than what is suggested by the test-time cost alone.
3. The method does not outperform RCP1 at small radii.
 The paper acknowledges that FLoR(1) can underperform RCP1 at very small perturbation radii. The explanation given is that finite-sample correction and the mismatch adjustment between tuning and validation (via \delta_0) introduce additional conservativeness.

Questions
1. For FLoR(1), can the authors provide a formal theoretical guarantee for the actual practical \lambda-selection procedure used in the experiments, rather than only for the idealized formulation?
2. For FLoR(k), how should practitioners choose the hyperparameters (k,\alpha_0,\beta_0) in a principled and computationally efficient way?

---

> ### Author Rebuttal · Authors · 2026-03-30
>
> We thank the reviewer for reading the paper, insightful questions, and positive assessment. We address all raised points below and commit to incorporating them in camera-ready.
>
> **W1/Q1.** Prop. 3.1 already provides a complete formal guarantee: any $\lambda$ satisfying the constraint is valid, and the finite-sample correction is handled by the Clopper–Pearson lower bound $\underline{\beta}$ with confidence level $\delta$. The supremum only finds the most efficient such $\lambda$.
>
> The naive approach is to run binary search over $\lambda$ as defined in Prop 3.1, re-sampling at each step — valid but expensive. Our tuning step is a cheap proxy for this search: it identifies a candidate $\lambda$ using far fewer samples (no statistical rigor is required for the search itself; therefore the same sample is used for the entire binary search), which is then validated exactly once against the constraint in Prop. 3.1. The statistical validity depends entirely on the final validation step, not on how the candidate was found. The idealized binary search via Prop. 3.1 and our tuning-then-validate pipeline provide identical formal guarantees; we simply circumvent the expensive optimization with a cheaper proxy. In practice, the tuning step took under 500 milliseconds across all CIFAR-10 runs, and in theory it replaces an expensive binary search block with the other, which in overall reduces the computation time.
>
> **W2/Q2.** The three hyperparameters $k$, $\alpha_0$, and $\beta_0$ are coupled through Eq. 5, but their selection is more efficient than it may appear:
>
> - **$k$** is a deployment-time design choice (e.g., batch size or concurrent inference budget). Higher $k$ is always better: the inner factor in Eq. 5 is monotonically increasing in $k$ for any $\beta_0 > 0$, which allows strictly smaller $\alpha_0$ and $\beta_0$.
> - **$\alpha_0$ and $\beta_0$**: fixing one determines the other via a binary search over Eq. 5. The equation is independent of the data, and computationally trivial -- in practice it takes few miliseconds.
> - In practice, tuning reduces to a one-dimensional search over $\alpha_0 \in (0, \alpha/2)$, with $\beta_0$ determined automatically. This can be performed on the same tuning set used for $\lambda$ selection, without any separate hold-out split.
> - As a practical default, setting $\alpha_0 = \alpha/10$ performs well across all our experiments.
>
> **W3.** Comparing RCP1 and FLoR(1) reduces to three factors. Both methods use the same prediction set structure; they differ only in how $1-\lambda$ is computed.
>
> 1. **Jensen's gap:** FLoR(1) estimates $E[c^\downarrow(\beta)]$ via Monte Carlo, while RCP1 uses $c^\downarrow(E[\beta])$. Since $c^\downarrow$ is convex, Jensen's inequality gives $E[c^\downarrow(\beta)] \geq c^\downarrow(E[\beta])$, so FLoR(1) obtains a strictly tighter threshold and thus smaller prediction sets. Via Taylor expansion: $E[c^\downarrow(\beta)] \approx c^\downarrow(E[\beta]) + \tfrac{1}{2} c^{\downarrow''}(E[\beta]) \operatorname{Var}[\beta]$. The second derivative of the Gaussian certified bound is 0 at $r = 0$, peaks at $r^\star = \frac{\Phi^{-1}(1-\alpha) + \sqrt{(\Phi^{-1}(1-\alpha))^2 + 4}}{2}$, and returns to 0 as $r \to \infty$, but is strictly positive for all intermediate $r$. Therefore $|C_{\mathrm{RCP1}}(x)| - |C_{\mathrm{FLoR}}(x)| \propto c^{\downarrow''}(E[\beta]) \operatorname{Var}[\beta] > 0$ for $r \in (0, r^\star)$.
>
> 2. **Finite-sample correction:** FLoR(1) calibrates to a higher target coverage and uses the Clopper–Pearson lower confidence bound, both of which increase conservativeness and inflate the set size. This effect diminishes with increasing sample rate, but for small radii the required sample rate is extremely high.
>
> 3. **Tuning conservativeness:** The tuning process introduces additional conservativeness to ensure a single certification run suffices, acting as the practical implementation bottleneck.
>
> Factor (1) is the sole source by which FLoR(1) outperforms RCP1; factors (2) and (3) work against it. For any fixed sample rate, there exists a threshold radius (very small for enough samples) below which RCP1 outperforms FLoR(1) (the Jensen gap is negligible there), and above which FLoR(1) always outperforms. For any fixed $r > 0$, there exists a sample rate above which FLoR(1) always outperforms RCP1, though for small radii this sample rate is impractically large.
>
> We thank the reviewer once again and commit to adding these clarifications in the camera-ready version.

---

> > ### Author Rebuttal · Reviewer_exuQ · 2026-04-01
> >
> > Thank the authors for the detailed responses. I would like to keep my score unchanged.
> > I have read the other reviewers’ comments and will keep track of the discussion between the authors and other reviewers.

---

### Decision · Program_Chairs · 2026-04-30

**Decision:**

Accept (regular)

**Comment:**

This paper proposes Front-Loaded Robust Conformal Prediction (FLoR), which increases Monte Carlo sampling at calibration while keeping test-time cost minimal (single or few samples), producing smaller prediction sets than the single-sample baseline RCP1 at moderate-to-large perturbation radii. A majority-vote variant, FLoR(k), further reduces prediction-set stochasticity.

All four reviewers converged on 4 (Weak Accept) with all concerns marked "Fully resolved." No correctness issues were raised. The main residual concern is limited novelty: two reviewers rate Originality at 2/4. Presentation was also rated poor (1/4) by two reviewers. In the AC-reviewer discussion, the one responding reviewer (UEhx) stated the contribution is "sufficient novel for ICML". From my perspective, this paper addresses a well-scoped practical gap, the guarantees are sound and the contribution is clean (while a bit incremental). I recommend acceptance.